# Deciphering cell-type-and temporally specific matrisome expression signatures in human cortical development and neurodevelopmental disorders via scRNA-seq meta-analysis

Do Hyeon Gim ⓘ, Muhammad Z. K. Assir ⓘ, Olivia Soper ⓘ, Paul A. Fowler ⓘ, Michael D. Morgan ⓘ, Daniel A. Berg ⓘ ✉ & Eunchai Kang ⓘ ✉

Human cortical development is a complex process involving the proliferation, differentiation, and migration of progenitor cells, all coordinated within a dynamic extracellular matrix (ECM). ECM plays a crucial role in guiding these processes, yet its specific contributions and the implications of its dysregulation in neurodevelopmental disorders (NDDs) remain underexplored. In this study, we conducted a meta-analysis of single-cell RNA sequencing (scRNA-seq) data from 37 donors, gestational weeks 8 to 26, across six independent studies to elucidate cell-type-specific matrisome gene expression signatures and their dynamics in the developing human cortex. Our analysis identified distinct matrisome gene signatures across various cell types, with significant temporal changes during cortical development. Notably, a substantial proportion of matrisome genes are associated with NDDs, exhibiting cell-type, temporal, and disease specificity. These findings highlight the critical role of cell-type-specific matrisome regulation in cortical development and its potential involvement in NDD pathogenesis. This study provides a comprehensive map of cell-type-specific matrisome signatures in the developing human cortex and highlights the importance of ECM in both normal development and the pathogenesis of NDDs.

Human cortical development involves a series of intricate processes, including the proliferation of progenitor cells, their differentiation into distinct cell types, and their subsequent migration and interconnection[1]. The precise coordination of these processes is crucial for shaping the cortex's developmental trajectory and defining its final architecture. While considerable progress has been made in understanding the regulation of individual processes, how they are coordinated through simultaneous interactions among different cell types remains poorly understood.

The extracellular matrix (ECM) is of particular interest because it provides a dynamic environment for interactions among various cell types. The ECM serves as a structural framework in which the cellular components of all tissues are embedded. In the developing human brain, the ECM constitutes approximately 40% of the total volume[2],

Institute of Medical Sciences, School of Medicine, Medical Sciences and Nutrition, University of Aberdeen, Aberdeen, UK. ✉e-mail: daniel.berg@abdn.ac.uk; eunchai.kang@abdn.ac.uk

offering crucial support and guidance for cellular organization and function, and regulating cell-cell communications by governing signaling pathways through the control of spread and restriction of signaling molecules[3,4]. During neural development, the ECM is essential for the proliferation and differentiation of neuronal progenitors[5–8], dendritic and axonal growth[9–12] and axonal guidance[12], neuronal migration[9], cortical folding[13–15], connectivity, and synaptic plasticity[16–18].

The importance of the ECM in cortical development is highlighted by brain malformations linked to ECM gene mutations, such as *RELN*, which is essential for neuronal migration and positioning[19,20]. Mutations in ECM-related genes, including *POMT1/2* and laminin subunit genes, *LAMB2*, and *LAMC3*, are associated with conditions such as cobblestone lissencephaly and polymicrogyria, leading to cortical layer disorganization and abnormal cortical folding[21–23]. Furthermore, mutations in ECM components like collagen are linked to malformations such as porencephaly (*COL4A1, COL4A2*) and Knobloch syndrome (*COL18A1*), where neuronal migration is disrupted[24].

While functional studies of ECM genes in animal models, particularly mice, have deepened our understanding of ECM roles during cortical development, significant differences exist between the ECM of the human and mouse cortex. The human fetal cortex ECM is more abundant and diverse than that of the mouse, especially rich in components such as hyaluronan, chondroitin sulfate proteoglycans, and other glycosaminoglycans[25,26]. These differences are also evident in their transcriptome profiles and gene expression patterns. In the developing human cortex at gestation weeks (GW) 13-16, the ECM transcriptome of the subventricular zone (SVZ) closely resembles that of the ventricular zone (VZ), suggesting a shared microenvironment that supports progenitor cell self-renewal, and includes distinct sets of collagens, laminins, proteoglycans, integrins, and specific growth factors[27]. In contrast, in the developing mouse cortex at E14.5, the SVZ ECM transcriptome more closely resembles that of the cortical plate (CP), indicating a fundamental difference in the organization and function of germinal zones across species[27]. Gene expression analysis further reveals that ECM-associated genes are highly expressed in both the VZ and SVZ in humans, suggesting a shared ECM environment that supports the self-renewal of neural stem cells (NSCs) and progenitors. In contrast, in mice, these genes are predominantly expressed in the VZ[27]. These differences might contribute to the greater plasticity and complexity of the human cortex.

The matrisome refers to the set of genes and proteins that compose and regulate the ECM[19]. To better understand the developmental processes and their dysregulation contributing to neurodevelopmental disorder (NDD) pathogenesis, it is important to systematically examine how the matrisome shapes human cortical development. It is particularly critical to understand how each cell type specifically contributes to the matrisome and mediates diverse biological processes throughout development.

Single-cell RNA sequencing (scRNA-seq) of human fetal brain tissue is a powerful tool that advances our understanding of cell-type-specific gene expression dynamics at single-cell resolution. This technology facilitates the analysis of cellular interactions across various developmental stages, providing detailed insights into the progression of cellular differentiation and development. By capturing snapshots of individual cells at different stages, scRNA-seq offers a comprehensive view of the cellular landscape, revealing the intricate processes underlying brain formation and maturation[28]. This method enhances our ability to study the complex interactions mediated by the matrisome and transitions that occur during neurodevelopment, contributing to a deeper understanding of brain function and the pathogenesis of NDDs.

However, the limited availability of human fetal tissue, due to ethical, legal, and logistical constraints, presents a significant challenge to using scRNA-seq to study dynamic brain development[28]. Consequently, individual studies often rely on a small number of available samples. These limitations prevent findings from fully capturing the diversity and complexity of cellular states and interactions present at each stage of dynamic fetal brain development, leading to gaps in our understanding of a comprehensive and continuous map of brain development.

Meta-analysis of scRNA-seq data from human fetal brain tissue offers substantial benefits in addressing the scarcity of tissue samples across different developmental stages. By integrating datasets from multiple studies, meta-analysis can significantly increase the sample size and reduce biases inherent in individual datasets, providing a more comprehensive overview of gene expression patterns and cellular dynamics throughout brain development[29]. Importantly, meta-analysis facilitates cross-validation of findings, enhancing the reliability and robustness of conclusions drawn from scRNA-seq data[29].

To elucidate cell-type-specific matrisome gene expression signatures and their dynamics across different developmental trajectories in the developing human cortex, we conducted a meta-analysis of scRNA-seq data encompassing GW 8 to 26 from six independent studies[30–35]. Our findings reveal that each cell type possesses unique matrisome gene expression signatures, which reflect the biological processes active during cortical development. These signatures undergo dynamic changes along specific differentiation lineages and throughout brain development. Additionally, we discovered that a substantial portion of matrisome genes is associated with NDDs exhibiting cell-type, temporal, and disease-specificity.

## Results

### Matrisome genes linked to NDDs through cross-referenced database analysis

The human matrisome consists of core matrisome proteins, including glycoproteins, proteoglycans, and collagens, as well as matrisome-associated proteins, which are classified into ECM-affiliated proteins, ECM regulators, and secreted factors that bind to the ECM (Fig. 1a). To date, 1,027 proteins have been identified in the human matrisome, comprising 274 core matrisome proteins and 753 matrisome-associated proteins (Fig. 1a)[36]. To investigate the association between ECM genes and NDD risk, we cross-referenced ECM genes with three NDD risk gene databases: the Simons Foundation Autism Research Initiative (SFARI) database, the Geisinger Developmental Brain Disorder Gene Database, and the Systems Biology of Neurodevelopmental Disorders (SysNDD) database[37–39]. These databases collectively identified 2,723 unique NDD risk genes, of which 139 are matrisome genes. We found that 17.2% of core matrisome genes and 9.8% of matrisome-associated genes are reported as NDD risk genes (Fig. 1b). Matrisome genes were identified as risk factors for various NDDs, including intellectual disability (ID), autism spectrum disorder (ASD), epilepsy (EP), attention deficit hyperactivity disorder (ADHD), schizophrenia (SCZ), and cerebral palsy (CP) (Fig. 1c). While some core matrisome NDD risk genes, such as *LAMA1, LAMA2, RELN, COL4A1, EYS, FBN2*, and *LAMB2*, were linked to multiple NDDs, the majority were associated with a single disorder (Fig. 1d). Similarly, matrisome-associated genes such as *F2, FGF13, FLG, NGLY1, SEMASA, CRLF1*, and *FGF14*, were found to be risk factors for more than one type of NDD (Fig. 1d). This finding indicates that both unique and shared matrisome genes are linked to NDDs, suggesting a potential role for their dysregulation in NDD development. Based on these results, we sought to uncover the cell-type-specific expression patterns of matrisome genes and their dynamic changes throughout human cortical development.

### A framework for scRNA-seq meta-analysis

To investigate cell-type-specific matrisome signatures during cortical development, we performed a comprehensive meta-analysis of scRNA-seq data from six independent studies encompassing 37 fetal cortex samples. The analysis pipeline consisted of three key steps: (1) raw count matrices were retrieved from all studies. (2), rigorous quality

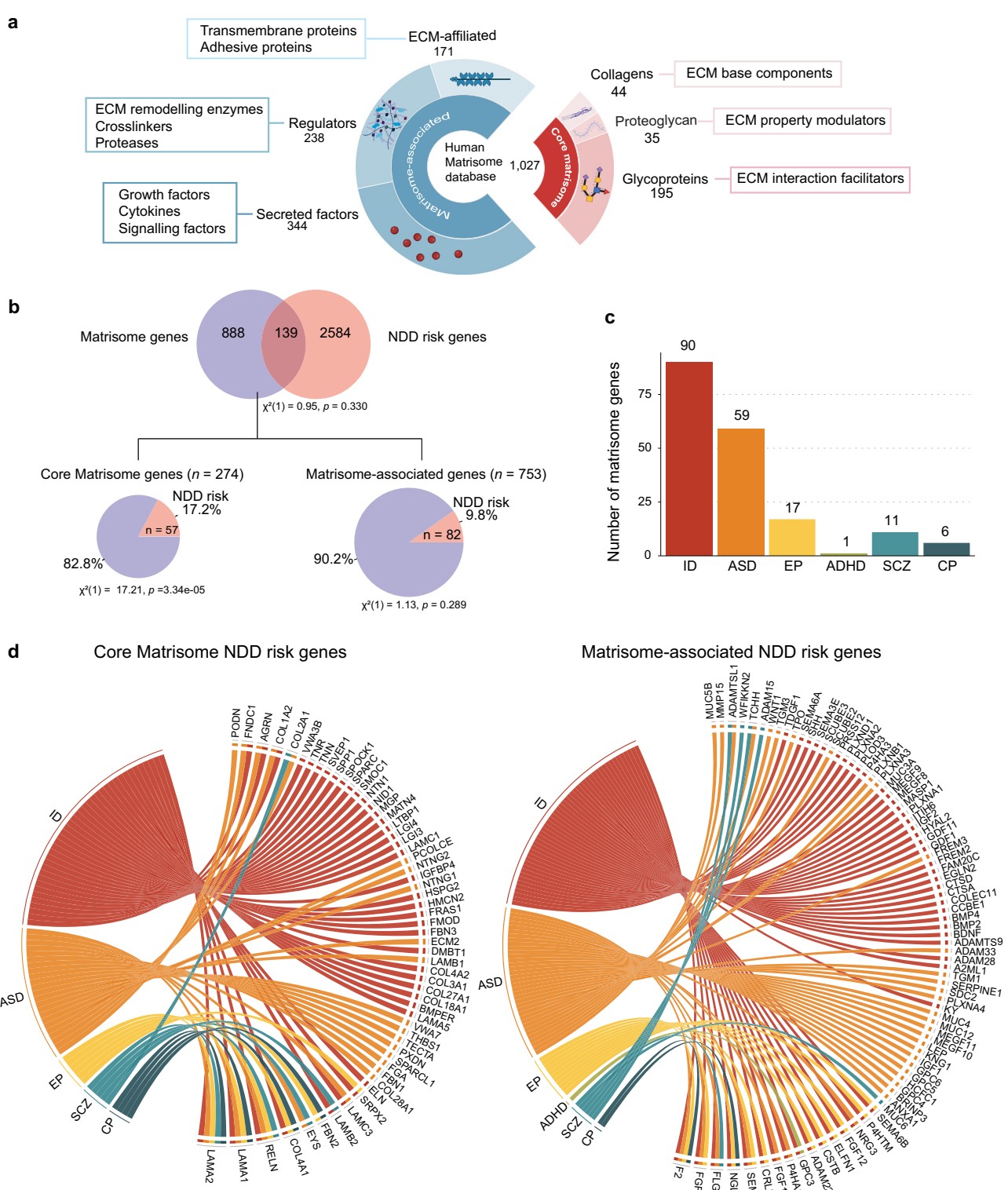

**Fig. 1 | Identification of NDD-Associated Matrisome Genes. a** Schematic overview of the different components of the core matrisome and matrisome-associated proteins. Created in BioRender. Gim, D. (2025) https://BioRender.com/6ddcyq5. **b** Venn diagram illustrating the overlap between matrisome genes and NDD risk genes. Pie charts represent the proportion of core matrisome and matrisome-associated genes that are classified as NDD risk genes. Chi-square tests of independence were performed for each comparison. **c** Number of matrisome genes identified as risk factors for each NDD. Intellectual disability (ID), autism spectrum disorder (ASD), epilepsy (EP), attention deficit hyperactivity disorder (ADHD), schizophrenia (SCZ), and cerebral palsy (CP). **d** Chord plots showing association between NDD types and their corresponding core matrisome (left) and matrisome-associated NDD risk genes (right).

control of each dataset was performed by removing low-quality cells and doubles, followed by normalization and log-transformation of gene counts. (3), the datasets were integrated using 2,000 anchor genes to create a unified meta-dataset (Fig. 2a). Subsequently, clustering on an integrated k-shared nearest neighbor (k-SNN) graph revealed 40 distinct clusters. Each cluster was annotated with a cell type label using a semi-supervised approach that combined the scType algorithm[40,41] with known cell type markers (Supplementary Fig. 1a and Supplementary Data 1). The final integrated dataset comprised 213,659 cells spanning GW 8 to 26, with varying contributions from each study (Fig. 2b). The successful integration of multiple datasets was visualized by the Uniform Manifold Approximation and Projection (UMAP), indicating minimal batch effects (Fig. 2c). We checked the data set integration for robustness using two measures: (1) the integration Local Inverse Simpson's Index (iLISI) and (2) cell-type LISI (cLISI) scores, which measure the batch-mixing and cell type grouping respectively[42]. In an ideal setting, iLISI should be high, while cLISI should be close to 1. Indeed, in our integrated data iLISI was 2.004 and cLISI was 1.277 (Fig. 2c, d). For comparison, we performed the same integration on a control dataset of human peripheral immune cells, achieving similar integration scores: an iLISI of 1.508 and a cLISI of 1.146 (Supplementary Fig. 1b). Additionally, we calculated the LISI score from the complete batch mixing of our meta-dataset, which was 2.714 (Supplementary Fig. 1c). These analyses further validated the robustness of our integration approach.

We annotated clusters into major cell types, including NSC, intermediate progenitor cells (IPC), immature glutamatergic excitatory neurons (Imm.GExN), maturing glutamatergic neurons (M.GExN), unspecific neurons, GABAergic inhibitory neurons (GABA IN), astrocytes, oligodendrocyte precursor cells (OPC), as well as non-neural cell types such as microglia and endothelial cells. This analysis also revealed distinct clusters of NSC subtypes including ventricular radial glia (vRG), outer radial glia (oRG), and truncated radial glia (tRG). (Fig. 2d, Supplementary Fig. 1a and Supplementary Data 2). The final annotated meta-dataset showed a balanced representation of major cell types, with each type being represented by multiple donors (Supplementary Fig. 1d).

We validated cell type annotations by examining the expression levels of canonical cell-type markers across annotated clusters (Fig. 2e) and inspecting the overlay of individual cell types along with the expression levels of their corresponding cell type markers on the integrated UMAP (Supplementary Fig. 2a). Notably, we identified multiple clusters concordant with vRG and oRG, indicating gene expression heterogeneity within these NSC populations (Supplementary Fig. 2b). We also identified a cluster of neurons that did not fit into known neuronal cell types (labeled as unspecific neuron). These neurons exhibited upregulated neuronal hemoglobin genes and increased metabolic activity, indicative of a less differentiated state, based on differential gene expression (DGE) analysis and subsequent gene ontology (GO) analyses (Supplementary Fig. 2c–e).

## Cell type-specific signatures of matrisome genes in the developing human cortex

We next sought to understand the cell type-specific and shared patterns of matrisome gene expression in our integrated meta-dataset. To identify potential patterns in matrisome gene expression across cell types, hierarchical clustering was applied to a matrix of average gene expression levels per cell type. This analysis revealed the presence of distinct matrisome gene expression signatures for each cell type among 953 detected matrisome genes (Supplementary Fig. 3).

To systematically identify matrisome marker genes across cell types, we performed DGE analysis using pseudobulk gene expression as input, comparing each cell type to all others. Pseudobulk aggregates gene expression data by cell type within each donor, correcting uneven cell sampling across studies and donors in the integrated meta-

dataset and enhancing the biological interpretability of the analysis by reducing cell-level noise (Fig. 3a). Most cell types exhibited distinct matrisome marker genes, with endothelial cells and astrocytes having the highest numbers (102 and 100, respectively), highlighting their prominent roles in ECM composition and remodeling as well as the propagation of signaling molecules (Fig. 3b). Microglia are known to interact with various cell types and are finely tuned to adapt to environmental cues, enabling them to adjust their roles in neuroprotection, immune defense, and tissue remodeling[43]. In agreement, the unique enrichment of matrisome-associated marker genes in microglia suggests their essential contribution to these functions.

To further assess the specificity of cell-type matrisome marker genes, the expression levels of up to the top five core matrisome and matrisome-associated marker genes for each cell type were visualized (Fig. 3c and Supplementary Data 3). The heatmaps revealed distinct matrisome gene signatures for each cell type. While some genes, such as *SPP1* in microglia and *SEMA* family genes in neuronal cells, are known to be enriched in these cell types, many remain unexplored in the context of cortical development, suggesting potential targets for future functional studies. Collagen, a key ECM component in the developing cortex, plays a crucial role in structural organization, neural stem cell behavior, neuronal migration, vascular development, and intercellular signaling[44]. Notably, our findings highlight that each cell type exhibits preferential expression of specific collagen subtypes (Fig. 3c). We observed that NSC subtypes share a common expression pattern of matrisome marker genes (Fig. 3c). To identify the distinct features of matrisome gene expression in each NSC subtype, we conducted additional DGE analysis comparing matrisome gene expressions among vRG, oRG, and tRG (Fig. 3d, Supplementary Data 4b). The differentially expressed matrisome genes in each NSC subtype reflect their unique functional roles in cortical development. In vRG, matrisome genes such as *RELN*, *COL1A2*, and *SMOC1*, which support ECM remodeling, neuronal migration, and structural organization[45–47], were upregulated. In oRG, *SPARCL1*, *LGI1*, *FBLN2* and *S100B*[48–50] showed relatively higher expression, contributing synaptic connectivity and organization, neurogenesis, and gliogenesis. In tRG, genes such as *NRG3*, *FGFBP2*, and *ITIH2* were enriched, facilitating ECM remodeling, signaling, and adaptive progenitor activity[51–53]. These distinct matrisome expression profiles suggest specialized functions of each NSC subtype during cortical development (Fig. 3d).

Collectively, our analyses identified unique, cell-type-specific matrisome signatures, suggesting the potential distinct roles of matrisome components in supporting each cell type's specialized functions during cortical development.

## Temporal dynamics of matrisome gene regulation during cortical development

Cortical development follows distinct temporal dynamics, characterized by shifts in cellular behavior, composition, and cytoarchitecture. NSCs transition from a proliferative to a neurogenic state, followed by gliogenesis, while the cortex undergoes layer formation, neuronal migration, and structural maturation, ultimately establishing its complex architecture[54].

To examine temporal changes in matrisome gene expression across all cell types during cortical development, particularly in NSCs, given their dynamic behavior throughout the developmental trajectory, we analyzed matrisome expression across three periods: late first trimester (GW 8–12), early-second trimester (GW 13–19), and late-second trimester (GW 20–26). These time windows align with critical transitions in cortical development, including progenitor expansion, neurogenesis, and early gliogenesis, respectively[1]. The number of donors for each cell type across age groups was generally comparable (Supplementary Fig. 4a). This strategy also enhances the robustness of temporal analyses by reducing noise from age-to-age variability while ensuring adequate representation of each cell type across

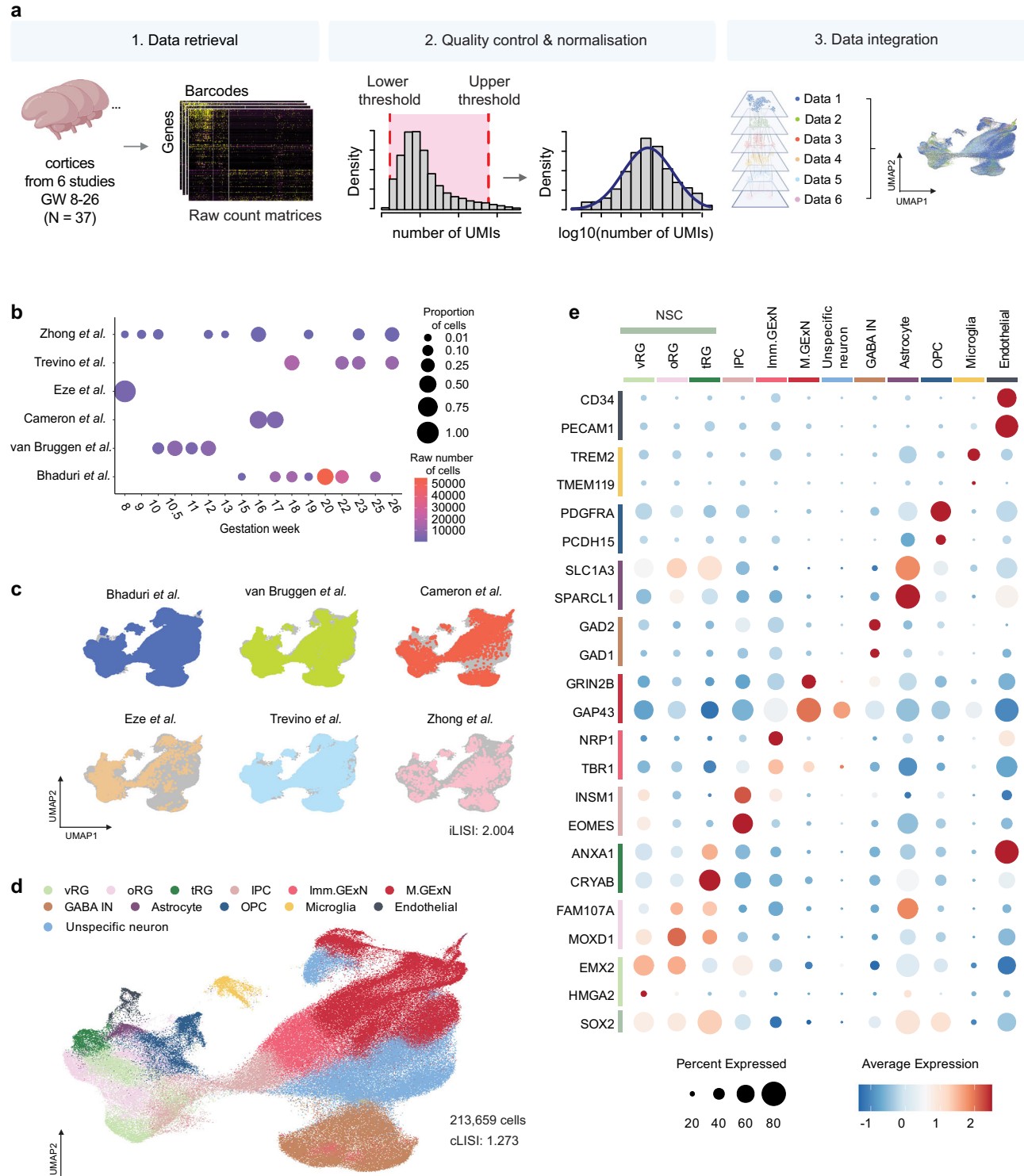

**Fig. 2 | Framework of scRNA-seq meta-analysis integrating multiple datasets.**
**a** Overview of the processing pipeline for scRNA-seq meta-data. Illustration created in BioRender. Gim, D. (2025) https://BioRender.com/z11x100. **b** Dot plot representation of each study's contribution to the meta-dataset across different gestational weeks. The proportion of cells is indicated by dot size, while the raw number of cells is represented by a color scale. **c** Distribution of scRNA-seq data from each study on the meta-data UMAP. The integration Local Inverse Simpson's Index (iLISI) for the meta-dataset is 2.004. **d** UMAP visualization of the integrated and clustered dataset, annotated and color-coded by 12 cell types. The cell type Local Inverse Simpson's Index (cLISI) for the meta-dataset is 1.273. **e** Dot plot representation of the expression of canonical markers for each cell type. Dot size indicates the percentage of cells expressing the gene, while dot color represents the average expression level.

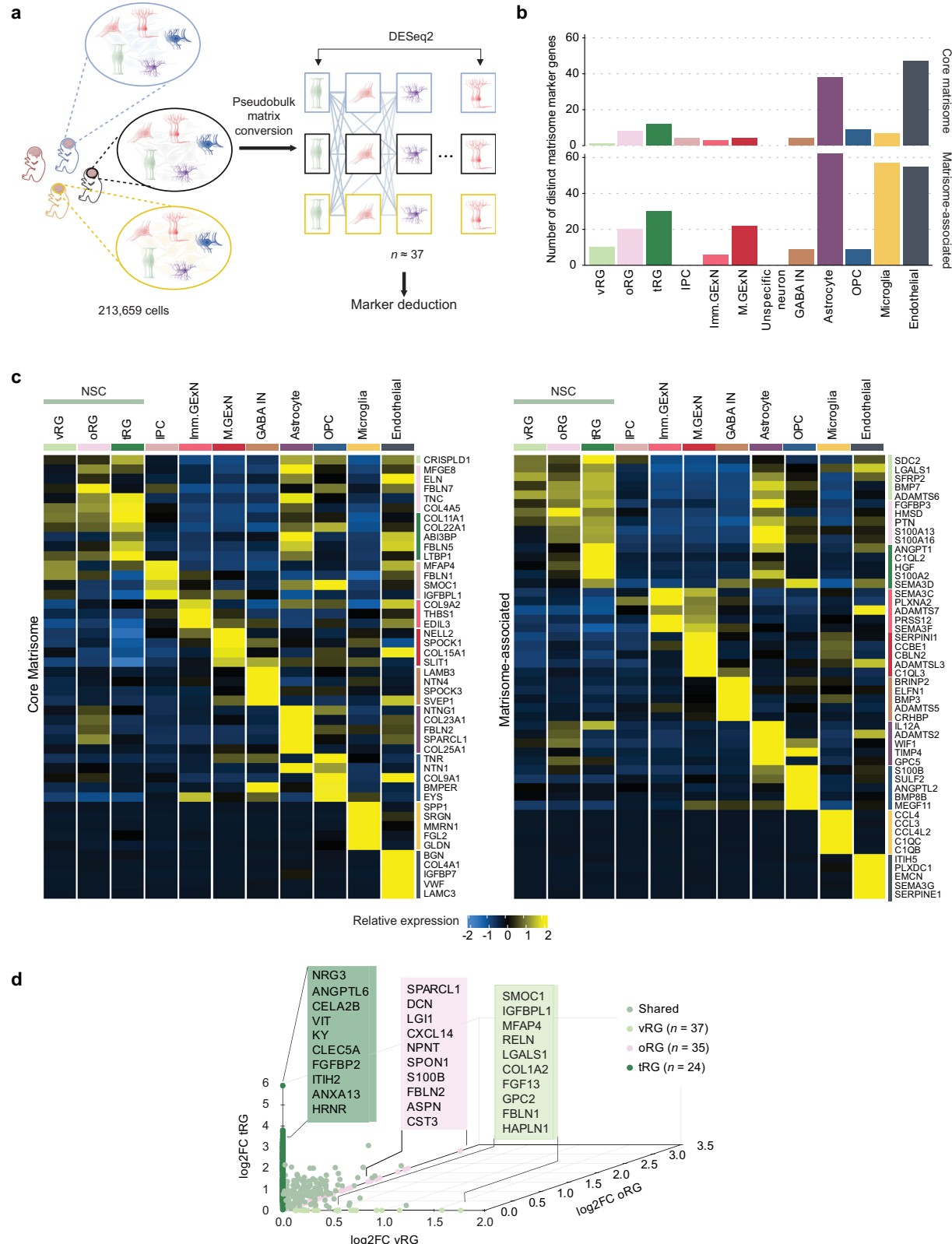

developmental periods. We used a linear model to evaluate the relationship between gene expressions in each cell type across all three developmental periods. The analysis revealed distinct temporal dynamics of matrisome gene expression across cell types in the developing human cortex (Fig. 4a and Supplementary Data 5). NSCs show notable increases in *TNC*, *LGALS3*, and *TIMP3*, while *FBLN1* exhibited a decreasing trend. Neuronal cell types demonstrate unique

matrisome signatures, with *MDK* showing a pronounced increase in IPC and M.GExN, and *PXDN* showing a temporal decrease in Imm.GExN and GABA IN. This signature is consistent with the previous understanding of *MDK*'s role in neural plasticity[55] and *PXDN*'s function in proliferation[56]. The analysis also revealed that all cell types display a higher number of temporally increasing matrisome genes than temporally decreasing genes (Supplementary Data 5). Non-neuronal

**Fig. 3 | Cell type-specific matrisome signature during cortical development.**
**a** Schematic representation of the pseudobulk approach, illustrating the aggregation of gene expression data by cell type for each donor. Created in BioRender. Gim, D. (2025) https://BioRender.com/h10f881. **b** Bar graph displaying the number of distinct core matrisome and matrisome-associated marker genes (log₂ fold-change > 1, adjusted *p*-value < 0.05) for each cell type. **c** Heatmap depicting the average expression levels of the top unique core matrisome and matrisome-associated marker genes for each cell type (ranked by log₂ fold change, adjusted *p*-value < 0.05). In both **c** and **b**, Differential expression was assessed using DESeq2

(negative binomial GLM, two-sided Wald test) with Benjamini–Hochberg FDR correction; effect size is log₂ fold-change with 95% Wald confidence intervals (*n* = as specified in Supplementary Fig. 1d). **d** 3D scatterplot of differentially expressed matrisome genes based on log₂ fold-change among NSC subtypes: ventricular radial glia (vRG, x-axis, pale green), outer radial glia (oRG, z-axis, pink), and truncated radial glia (tRG, y-axis, dark green). The number of donors per cell type is indicated. Matrisome genes that are highly expressed in at least two NSC subtypes are highlighted in pastel green.

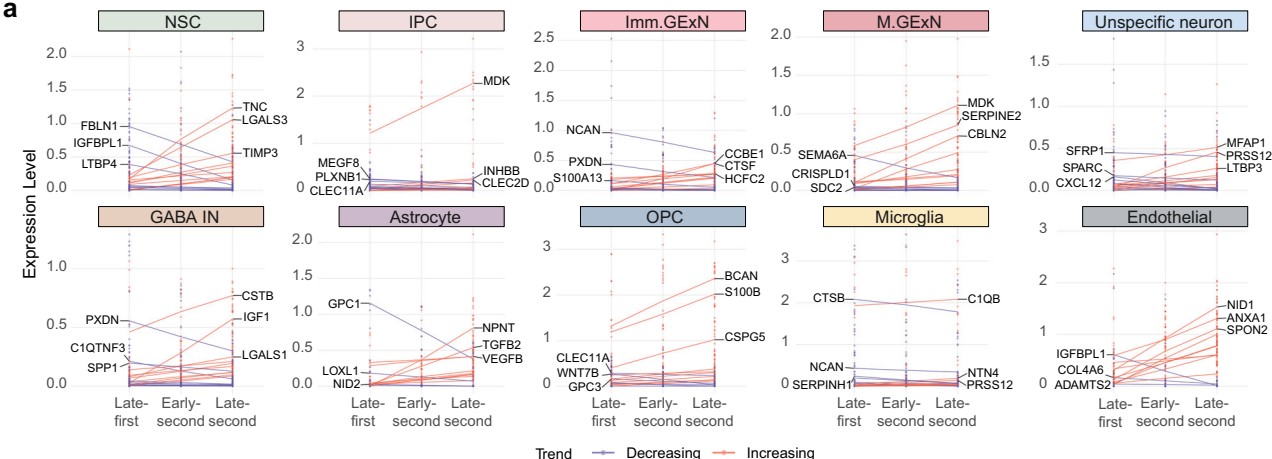

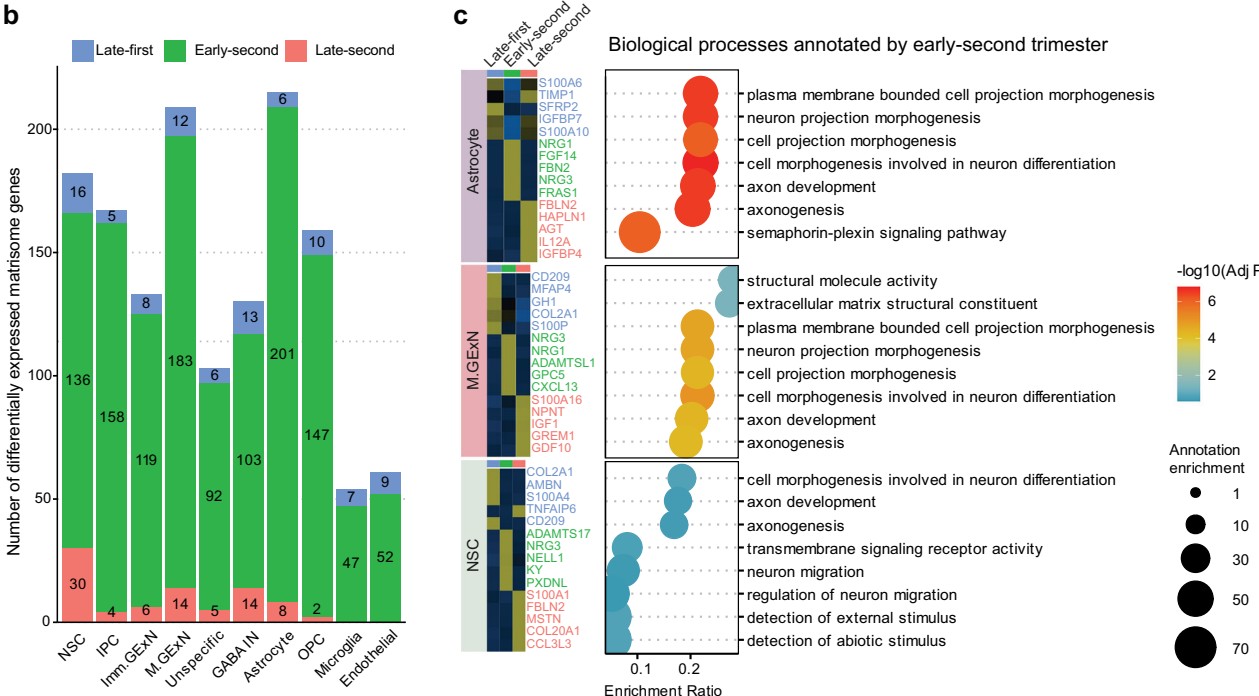

**Fig. 4 | Temporal dynamic signatures of matrisome genes in each cell type during cortical development. a** Temporal expression patterns of matrisome genes across distinct cell types during human cortical development. Each data point represents the expression level of matrisome genes in an individual donor, with a best-fitted trend line illustrating the expression pattern. Expression levels of the top 10 matrisome genes showing temporal increases (red) and decreases (blue) are displayed, and the three most significant genes are labeled for each cell type across three developmental periods. **b** Bar graph displaying the number of differentially expressed matrisome genes in each cell type across three developmental periods (log₂ fold-change > 1, adjusted *p*-value < 0.05). **c** On the left, a heatmap

displaying the expression levels of up to the top five matrisome genes across three developmental periods in astrocyte, GExN, and NSC (ranked by log₂ fold-change, adjusted *p*-value < 0.05). On the right, GO enrichment analysis of early-second trimester marker genes (log₂ fold-change>1, adjusted *p*-value < 0.05) against a matrisome gene background in astrocyte, GExN, and NSC. The x-axis represents the enrichment score, dot size indicates the number of genes in the annotation, and the -log₁₀ adjusted *p*-value is represented by the color scale. In **b** and **c** (left), differential expression was assessed using DESeq2 (negative binomial GLM, two-sided Wald test) with Benjamini–Hochberg FDR correction; effect size is log₂ fold-change with 95% Wald confidence intervals (*n* = as specified in Supplementary Fig. 1d).

lineage cell types exhibit the strongest tendency toward positive temporal regulation, with OPCs displaying 80.6% (25 out of 31 temporally regulated genes) and endothelial cells showing an 85.0% (17 out of 20 temporally regulated genes) of temporally increasing matrisome genes. For example, temporally upregulated genes in endothelial cells include *COL4A6, PRELP, PCOLCE,* and *NID1,* which are known to contribute to vascular basement membrane assembly[57–60]. These temporal dynamics of matrisome genes suggest their growing contribution to matrisome-mediated functions during cortical development (Supplementary Data 5).

Next, we sought to identify matrisome gene expression patterns specific to a particular development period in each cell type. We analyzed cell-type-specific temporal matrisome expression signatures using DGE analysis across three developmental periods. The early-second trimester exhibited a significantly higher number of differentially expressed matrisome marker genes (Fig. 4b and Supplementary Data 6). This reflects robust matrisome activity during this stage, aligning with the diverse cellular processes occurring at this period such as active neurogenesis, radial glial scaffold maturation, emergence of oRG and tRG, neuronal migration, and the early stages of gliogenesis and cortical lamination[1]. We visualized up to the top five matrisome marker genes for each developmental stage in each cell type, revealing distinct and stage-specific temporal matrisome expression signatures (Supplementary Fig. 4 and Supplementary Data 6). Astrocytes, M.GExN, and NSCs had the highest number of temporal marker genes, suggesting their dynamic matrisome gene expression patterns during development. To explore the functions of their matrisome genes during the early-second trimester, GO enrichment analysis was performed on the early-second trimester matrisome marker genes for those cell types. In astrocytes and M.GExN, genes associated with cell morphogenesis, including *FBN2, FRAS1, ADAMTSL1,* and *GPC5,* as well as genes involved in axon development, such as *NRG1, NRG3,* and *FGF14,* were enriched. In NSCs, genes involved in axon development, neuronal migration and maturation, including *NRG3, ADAMTS17, NELL1, KY,* and *PXDNL* were enriched (Fig. 4c). These represent important functions of temporally fine-tuned expression of matrisome genes in each cell type during development. In particular, the enriched expression of matrisome genes associated with axon development, cellular morphogenesis, and neuronal maturation and migration in astrocytes and NSCs highlights the importance of cell-cell interactions (CCI) mediated by the matrisome.

## Matrisome-mediated cell-cell communication during cortical development

To examine the role of matrisome genes in cell communication, we analyzed CCI and signaling pathways mediated by matrisome genes using CellChat[61]. Analysis of cell type-specific interaction networks revealed differences between whole-transcriptome-wide and matrisome-specific communication in the developing cortex. The whole transcriptome-based network showed dense interconnectivity across all cell types, with strong interactions primarily involving glial cells (vRG, oRG, tRG, OPC, and astrocytes) and endothelial cells. In contrast, the matrisome-specific network exhibited selective connectivity, with tRG, astrocytes, and endothelial cells emerging as major contributors to CCI. Notably, interactions among tRG, astrocytes, and endothelial cells in the whole transcriptome heavily rely on matrisome genes. Additionally, autocrine interactions mediated by matrisome genes were exclusively observed in tRG and astrocytes (Fig. 5a). These analyses indicate that the matrisome may play a role in communication between specific cell types during cortical development.

Signaling pathway-specific analysis identified three matrisome-mediated signaling networks during cortical development: the pleiotrophin (PTN), midkine (MK), and semaphorin 6 A (SEMA6) pathways (Fig. 5b). The PTN pathway, known for its role in modulating

proliferation, differentiation, and cell survival[62,63], exhibited robust bidirectional communication between neural and glial cells, with signaling notably converging on tRG and astrocytes. The MK signaling network, which supports cell growth and neural plasticity[64], showed a more focused interaction pattern, primarily targeting tRG. In contrast, SEMA6 signaling, associated with neuronal morphogenesis and migration[65,66], revealed distinct interactions targeting neuronal populations. Notably, tRG, astrocytes, and endothelial cells were identified as source cell types for all three pathways, highlighting their active roles in coordinating cell communication during cortical development (Fig. 5b).

Quantitative analysis of ligand-receptor pairs identified PTN-SDC3, MDK-SDC2, and SEMA6A-PLXNA4 as the key contributors within their respective signaling pathways (Fig. 5c). This analysis also uncovered cell type-specific communication patterns, with significant interactions observed between distinct cellular populations. Notable examples include strong PTN-SDC2 and PTN-SDC3 signaling from vRG/oRG to tRG and astrocytes, respectively, as well as SEMA6A-PLXNA2/ signaling from tRG/astrocytes/endothelial cells to Imm.GExN and M.GExN (Fig. 5d). PTN-SDC3 signaling induces neurite growth and migration of target cells[67] while SEMA6A-PLXNA4 signaling results in termination of neuronal migration[68]. This indicates potential roles of vRG/oRG in promoting migration of astrocytes and tRG/astrocytes/endothelial cells in the termination of migration of Imm.GExN and M.GExN.

Together, these findings highlight the matrisome's specialized role in mediating cell-cell communication during cortical development, with tRG, astrocytes, and endothelial cells emerging as key potential regulators of these interactions.

## Characterization of neuronal lineage-specific changes in matrisome gene expression

NSCs exhibit substantial heterogeneity during cortical development, shaped by their diverse fate choices along distinct developmental trajectories[35]. To explore matrisome signatures during neurogenesis, we performed DGE analysis to compare NSCs, IPCs, and excitatory neurons (GExN, including Imm. GExN and M.GExN). This analysis revealed distinct changes in matrisome signatures along the trajectory of the neurogenic lineage (Fig. 6a and Supplementary Data 7). Among the differentially expressed matrisome genes, *LGALS3* stood out due to its pronounced temporal dynamics in NSCs (Fig. 4a) and a significant expression difference between NSCs and GExN (log2 fold-change > 4). *LGALS3* encodes β-galactoside-binding Galectin-3 (GAL3). Indeed, our immunofluorescence staining revealed GAL3 expression in SOX2+ RG, but not in TBR2+ IPC (Supplementary Fig. 5a). GAL3 has been suggested as a transcriptional marker for oRG through scRNA-seq[69], but its validation has not been previously reported. Consistent with these findings, weighted gene co-expression network analysis (WGCNA) and Pearson's correlation analysis revealed that *LGALS3* is co-expressed with *HOPX,* a well-established oRG marker (Fig. 6b, c). To validate the expression of *LGALS3* and *HOPX,* we performed immunofluorescent co-staining of GAL3 and HOPX in human fetal prefrontal cortices at GW 16 and GW 17. 71.1 ± 4.2% of HOPX+ cells expressed GAL3 (Fig. 6d, e). The proportion of cells with high *LGALS3* expression increased during the early-second trimester (GW13–19), which was confirmed by immunofluorescence staining (Fig. 6f, g).

While oRG shows strong *HOPX* expression, vRG exhibits minimal expression[69]. To examine the potential role of *LGALS3* in oRG, we further identified *LGALS3*+ cells with high *HOPX* expression (*HOPX*high). More than 44% of *LGALS3*+*HOPX*high were annotated as oRG. Additionally, a significant proportion of *LGALS3*+*HOPX*high were annotated as macroglia, including astrocytes and OPCs, potentially indicating their differentiation into macroglia (Fig. 6h). To explore the developmental positioning of *LGALS3*+*HOPX*high cells along a differentiation trajectory, we performed pseudotime analysis with vRG as the root cell type

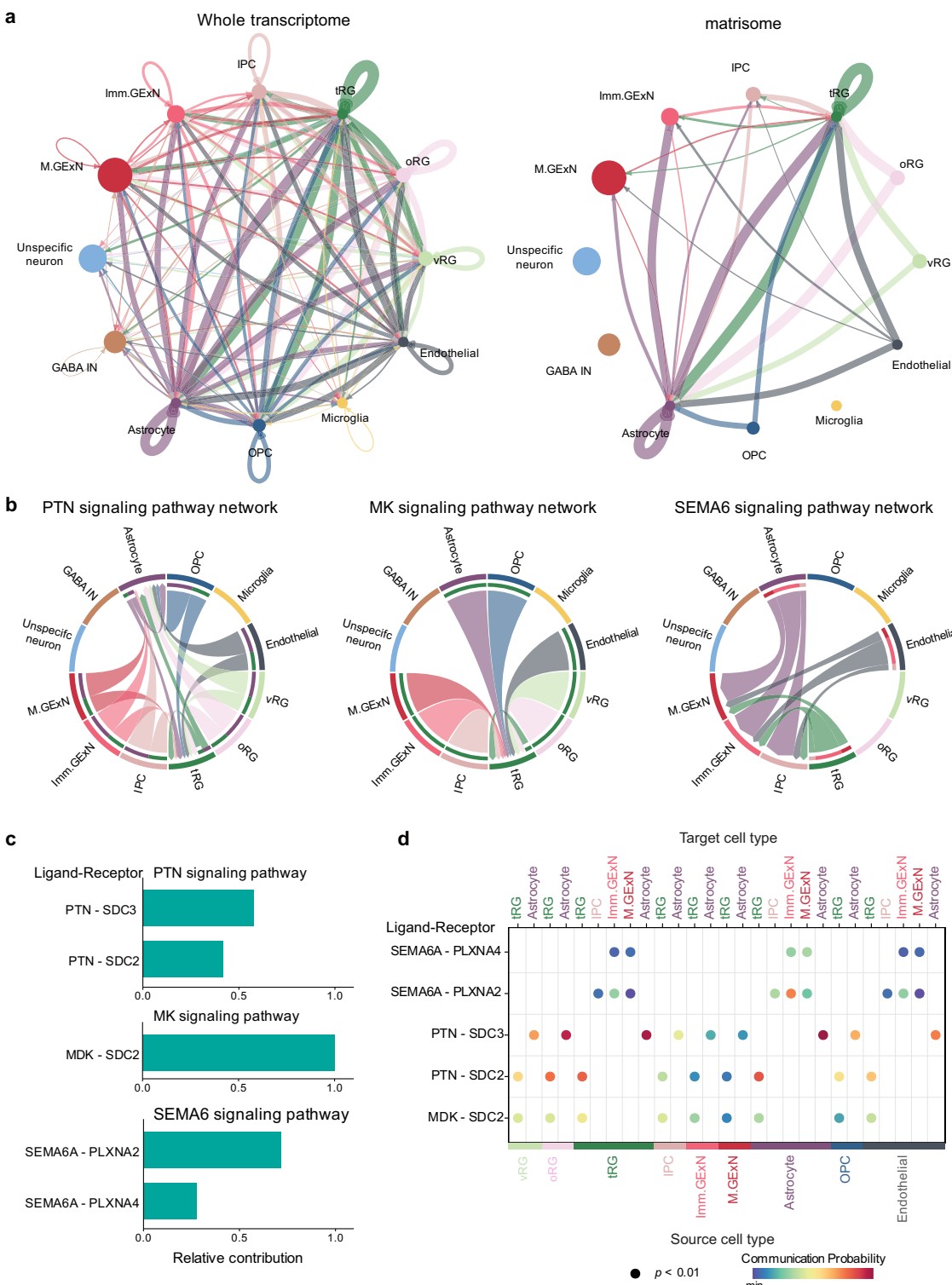

**Fig. 5 | Cell-cell communication mediated by matrisome. a** Network visualization of interaction weights/strength between cell types based on whole transcriptome (left) and matrisome (right) gene expression. Node size represents the relative abundance of cell types in the meta-data, while edge thickness indicates interaction strength. Arrow direction denotes the flow of signaling from sender to receiver. **b** Chord plots illustrating cell-type-specific signaling networks for PTN, MK, and SEMA6 pathways. Arc width represents the interaction strength between cell types. **c** Bar graph showing the relative contribution of individual ligand-receptor pairs to overall pathway activity. Bar length represents the normalized contribution strength. **d** Dot plot illustrating source-target cell type relationships for each ligand-receptor pair obtained from permutation analysis in CellChat. Dot size represents statistical significance ($p < 0.01$), while color intensity indicates communication probability.

(Supplementary Fig. 5b). The analysis showed that *LGALS3⁺HOPXʰⁱᵍʰ* cells were assigned a higher pseudotime value, consistent with a more mature differentiation state compared to vRG, oRG, tRG, and IPC (Fig. 6i). To understand the transcriptional similarities among

*LGALS3⁺HOPXʰⁱᵍʰ* cells and other cell types, we performed principal component analysis (PCA) on the pseudobulk gene expression profile. Prior to the analysis, we randomly assigned cells into three arbitrary groups and aggregated gene expression data to mitigate potential

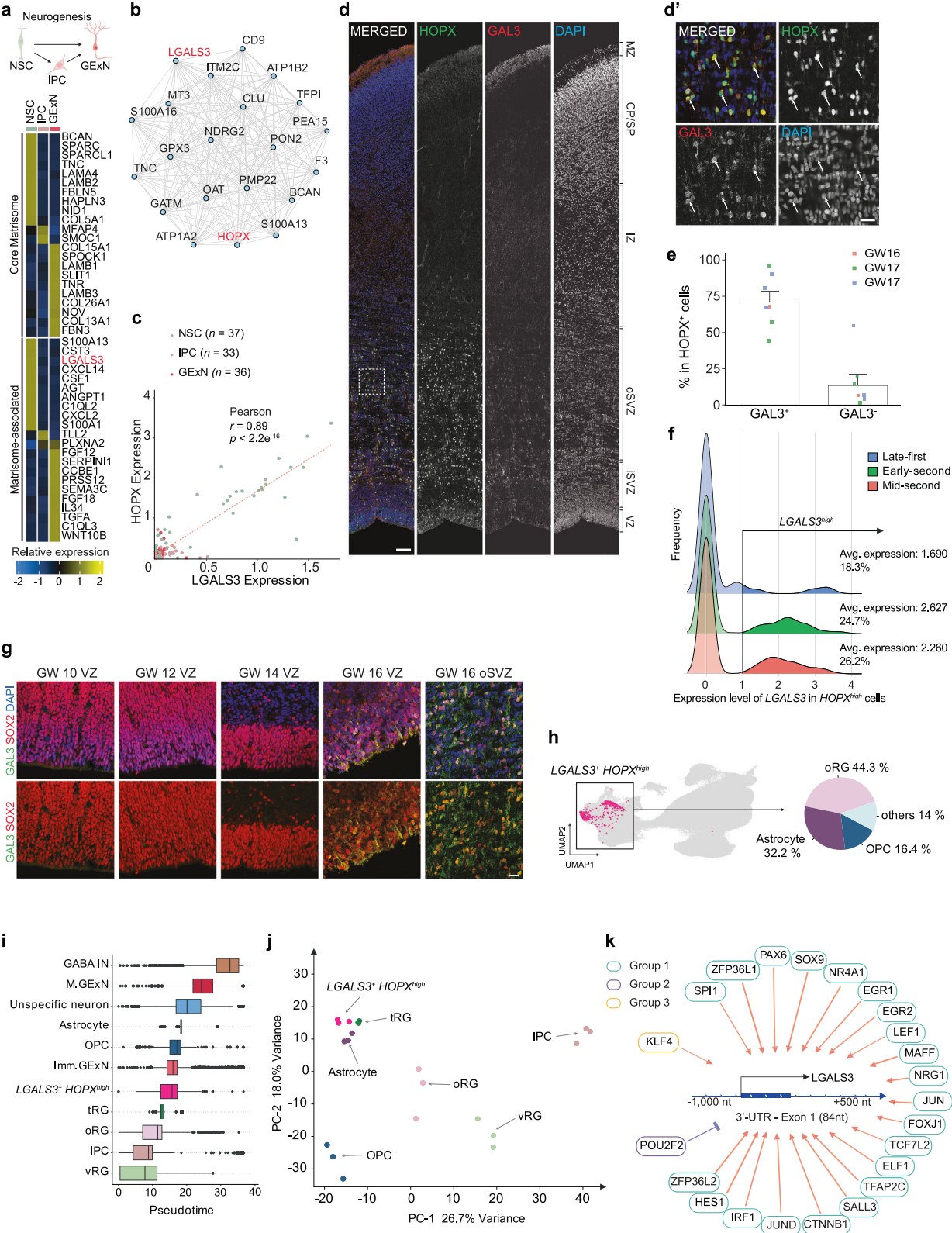

noise from gestational variability among donors and the large cell numbers in each cell type in our integrated dataset. PCA showed *LGALS3⁺HOPXʰⁱᵍʰ* cells clustering near astrocyte and tRG, distinct from IPC and OPC (Fig. 6j). Our analyses suggest that *LGALS3⁺HOPXʰⁱᵍʰ* cells may represent a subpopulation of NSCs with a preferential lineage trajectory towards astrocytes.

To further examine the transcriptional regulators of *LGALS3⁺* cells, we conducted an integrative meta-analysis utilizing ATAC-seq and ChIP-seq datasets from the ChIP-Atlas database[70–72]. Our analysis revealed that *LGALS3* promoter region and transcription start site (TSS) are predominantly in an open chromatin state in the human frontal cortex (Supplementary Fig. 5c). Notably, H3K27me3 marks,

**Fig. 6 | Characterization of neuronal lineage-specific changes in matrisome gene expression. a** Illustration depicting the neurogenic lineage, Created in BioRender. Gim, D. (2025) https://BioRender.com/k51v991. Heatmap displaying the expression levels of up to the top 10 cell-type-specific core matrisome (top) and matrisome-associated (bottom) marker genes across cells in the neurogenic lineage (ranked by $\log_2$ fold-change, adjusted $p$-value < 0.05). Differential expression was assessed using DESeq2 (negative binomial GLM, two-sided Wald test) with Benjamini–Hochberg FDR correction; effect size is $\log_2$ fold-change with 95% Wald confidence intervals ($n$ = as specified in Supplementary Fig. 1d). **b** Co-expression network of *LGALS3* across NSC, IPC, and GExN, highlighting the top significantly co-expressed genes identified through weighted gene co-expression network analysis. **c** Scatter plot showing the correlation between *LGALS3* and *HOPX* in scRNA-seq meta-data. The two-tailed Pearson correlation coefficient ($r$) and $p$-value for the best-fitted line are displayed. Each colored outline represents a donor per cell type. $df$ = 104 and 95% CI [0.846, 0.926]. **d-d'** Immunofluorescence staining of HOPX, Galectin-3 (GAL3), and DAPI in the GW16 fetal prefrontal cortex. The inset (d') provides a magnified view in oSVZ. Scale bars: 100 µm (d) and 10 µm (d'). Note the immunoreactivity of GAL3 in the marginal zone, which contains LGALS3-expressing microglia and radial glial end feet, with additional signals potentially arising from extracellular matrix localization and neighboring sources such as meningeal macrophages, fibroblasts, and myeloid-derived cells. The experiment was repeated 5 times independently with similar results. **e** Quantification of GAL3⁺HOPX⁺ and GAL3⁻HOPX⁺ cells as a percentage of the total HOPX⁺ cell population from immunofluorescence staining in human fetal prefrontal cortices at GW 16–17. Data is from three donors and presented as mean ± SEM. Source data are provided as a Source Data file. **f** Density plots illustrating the frequency distribution of *HOPX^high* NSCs with LGALS3 expression across human fetal developmental stages. *HOPX^high* is defined as expression level > average *HOPX* expression level in NSC and *LGALS3^high* is defined as expression level >1, as indicated on the horizontal axis. **g** Immunofluorescence staining of SOX2, GAL3 and DAPI in the fetal prefrontal cortex VZ at GW 10, 12, 14 and 16, and in the oSVZ at GW 16. Scale bars: 20µm. The experiment was repeated 3 times independently with similar results. **h** UMAP of scRNA-seq meta-data highlighting *LGALS3⁺HOPX^high* cells (magenta) with a pie chart showing the cellular composition of *LGALS3⁺HOPX^high* population. *HOPX^high* is defined as expression level > average *HOPX* expression level in NSC and *LGALS3⁺* is defined as expression level > 0. **i** Box plots illustrating the developmental trajectory across different cell types in the scRNA-seq meta-data. The x-axis represents pseudotime values, with boxes indicating the median and quartiles of each population's distribution. Cell populations are arranged in descending order from the top based on their mean pseudotime. In all box plots, the center line represents the median, the box bounds correspond to the 25th and 75th percentiles, and whiskers extend to the minimum and maximum values, and any points beyond the whiskers are defined as outliers. **j** Principal component analysis plot illustrating the transcriptional relationships between different cell populations. **k** Schematic representation of the *LGALS3* regulatory region and its associated TFs. The diagram illustrates the genomic region spanning -1,000 to +500 nucleotides relative to the *LGALS3* transcription start site. Red arrows indicate transcriptional activation, while blue blunt arrow represents transcriptional inhibition. Each group represents a module of TFs with a shared expression pattern over pseudotime.

enriched in neural progenitor cells, suggest a potential primed transcriptional state, while astrocytes exhibit H3K27ac marks, indicating active transcription (Supplementary Fig. 5c)[73,74]. We next performed integrated regulatory network analysis (IReNA) to identify transcription factors (TFs) that bind to the promoter regions of *LGALS3* potentially regulating its expression[75]. Twenty-three TFs were identified and grouped by shared expression patterns along the developmental trajectory, exhibiting either positive or negative co-expression with *LGALS3*. Importantly, the genomic region spanning 1,000 nucleotides upstream to 500 downstream of *LGALS3*'s TSS contains binding motifs for these TFs, suggesting potential direct regulatory roles (Fig. 6k and Supplementary Data 8). GO analysis revealed that transcriptional activators of *LGALS3* are associated with gliogenesis and glial cell differentiation (Supplementary Fig. 5d). Collectively, these findings suggest an astrocytic developmental trajectory for *LGALS3⁺* cells.

## Characterization of macroglial lineage-specific changes in matrisome gene expression

Gliogenesis, alongside neurogenesis, is a key lineage specification process in the developing cortex. To investigate matrisome gene expression dynamics during gliogenic lineage progression, we analyzed the matrisome signatures of NSCs, astrocytes, and OPCs using DGE analysis. The heatmap revealed distinct expression patterns of core and matrisome-associated marker genes across NSC, astrocyte, and OPC populations (Fig. 7a and Supplementary Data 9). Astrocytes and OPCs exhibited a broader range of matrisome markers than NSCs. Notably, *S100B*, a well-established astrocyte marker, was enriched in OPCs. While *S100B* is known to be expressed in OPCs and immature oligodendrocytes in the developing mouse brain[76,77], its expression dynamics in OPCs during human cortical development remain poorly understood.

WGCNA revealed significant co-expression of *S100B* with *PDGFRA*, *OLIG1*, and *OLIG2* (Fig. 7b). Moreover, a strong correlation between *OLIG2* and *S100B* expression in OPCs was confirmed by Pearson's correlation analysis (coefficient: 0.61, Fig. 7c), with 53.6% of *OLIG2^high* cells expressing *S100B* (Fig. 7d). To validate this, immunofluorescence staining of S100β and OLIG2 was performed in human fetal prefrontal cortices at GW 15 and GW 17, revealing that 74.8 ± 6.4% of OLIG2⁺ cells are S100β⁺ (Fig. 7e, f). To assess *S100B* expression

patterns in the context of fate specification, we performed a comparative analysis of mean expression levels of each donor for NSCs, astrocytes and OPCs, revealing that OPCs exhibit significantly higher *S100B* expression levels than NSCs or astrocytes (Fig. 7g). Collectively, our analyses identify the strong expression of *S100B* in OPCs during human cortical development.

## Cell type-and temporally specific expression of matrisome genes associated with NDDs

A fundamental approach to understanding the pathogenesis of NDDs is to study the function of NDD risk genes. A major challenge lies in examining gene function within the appropriate cell- type and developmental context. Through our analyses, we identified cell-type-and temporally specific signatures of matrisome genes in the developing cortex (Figs. 3, 4). Accordingly, we sought to characterize cell-type-specific and time windows of NDD risk matrisome gene expression, providing insights for the design of functional studies of risk genes. First, we quantified the number of cell-type-specific matrisome marker genes associated with specific NDDs across different cell types. The analysis revealed that matrisome marker genes of tRG, astrocytes, OPCs, and endothelial cells show the highest number of matrisome marker genes associated with NDDs (Fig. 8a and Supplementary Data 10). ID exhibited the highest number of enriched matrisome marker genes across all cell types, with a particularly strong representation in endothelial cells. In contrast, matrisome marker genes associated with ASD were predominantly enriched in non-neural cells. To assess the specificity of cell-type matrisome marker genes associated with NDDs, the expression levels of the top matrisome marker genes were analyzed and visualized in volcano plots and a heatmap (Fig. 8b, c, Supplementary Fig. 6 and Supplementary Data 11). Some NDD risk matrisome genes exhibited high cell-type specificity, with well-characterized functional significance. For example, *SPP1*, an ID risk gene, is highly expressed in microglia and was recently identified as a key regulator of structural integrity during brain development in mice[78]. *COL4A1*, predominantly expressed in endothelial cells, is linked to multiple NDDs, including ID, EP, CP, and neonatal hemorrhages, and plays a crucial role in human fetal vascular development[79] (Fig. 8c).

Next, we investigated the temporal specificity of NDD risk matrisome genes in each cell type using DGE analysis. All cortical cell types exhibited temporally distinct NDD risk matrisome genes, particularly

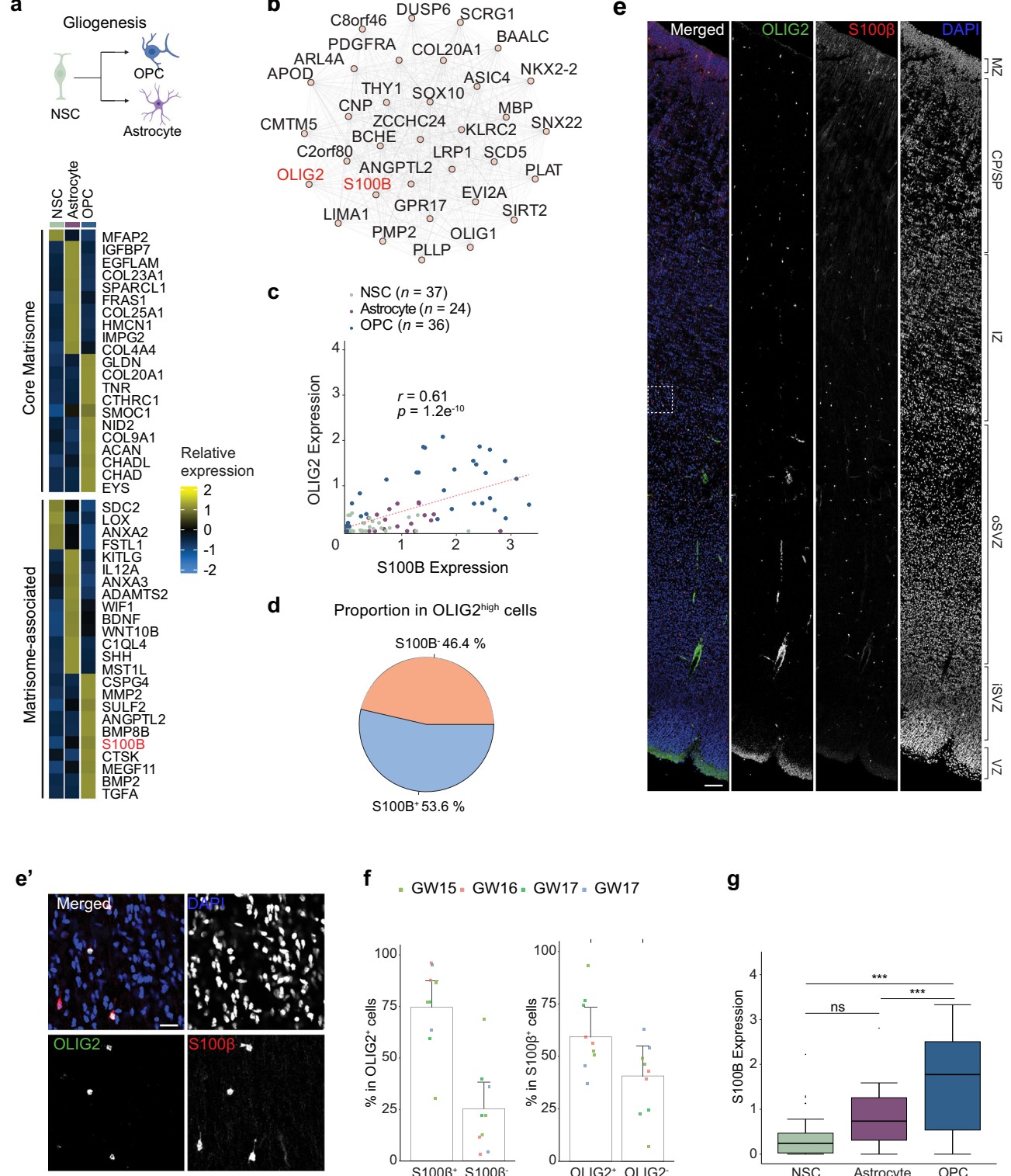

during the late first and early-second trimester (Fig. 8d and Supplementary Data 12). Consistent with our analysis of the temporal dynamics of matrisome genes (Fig. 4), many NDD risk matrisome genes were predominantly enriched in the early-second trimester. Notably, *NRG3* was enriched across all cell types during this period, aligning with its expanding roles in glial cell growth, excitatory synapse development, and neural plasticity[80,81].

Together, our analyses identified cell-type-and temporally specific NDD-associated matrisome genes. These findings not only highlight

the importance of matrisome gene regulation in cortical development and NDD pathogenesis but also provide insights for future functional studies.

## Discussion

The ECM plays a pivotal role in cortical development, acting as a structural scaffold that supports cellular organization, migration, and differentiation[3,82]. Despite its fundamental importance, the ECM's specific contributions to cortical development and its implications for

**Fig. 7 | Characterization of macroglial lineage specific changes in matrisome gene expression. a** Illustration depicting the macroglial lineage, created in BioRender. Gim, D. (2025) https://BioRender.com/k51v991. Heatmap displaying the expression levels of up to the top 10 cell-type-specific core matrisome (top) and matrisome-associated (bottom) marker genes across cells in the macroglial lineage (ranked by $\log_2$ fold-change, adjusted $p$-value < 0.05). Differential expression was assessed using DESeq2 (negative binomial GLM, two-sided Wald test) with Benjamini−Hochberg FDR correction; effect size is $\log_2$ fold-change with 95% Wald confidence intervals ($n$ = as specified in Supplementary Fig. 1d). **b** Co-expression network of *OLIG2* across NSC, astrocyte, and OPC, highlighting the top significantly co-expressed genes identified through weighted gene co-expression network analysis (WGCNA). **c** Scatter plot showing the correlation between *S100B* and *OLIG2* in scRNA-seq meta-data. The two-tailed Pearson correlation coefficient and $p$-value for the best-fitted line are displayed. Each colored outline represents a donor per cell type. df = 90 and 95% CI [0.462, 0.724]. **d** Pie chart showing the proportion of *S100B*⁺ cells among *OLIG2*^high^ cells in the scRNA-seq meta-data. *OLIG2*^high^ is defined as expression level > average *OLIG2* expression level in OPC and *S100B*⁺ is defined as expression level > 0. **e-e'** Immunofluorescence staining of OLIG2, S100B, and DAPI in the GW 16 fetal prefrontal cortex. The inset provides a magnified view. Scale bars: 100 μm and 10 μm (inset). **f** Quantification of S100β⁺ OLIG2⁺ and S100β⁻ OLIG2⁺ cells as a percentage of the total OLIG2⁺ cell population (left) and S100β⁺ OLIG2⁺ and S100β⁺ OLIG2⁻ cells as a percentage of the total S100β⁺ (right) from immunofluorescence staining in human fetal prefrontal cortices at GW 15–17. All data are from four donors and presented as mean ± SEM. Source data are provided as a Source Data file. **g** Box plots showing the mean expression levels of *S100B* in NSCs ($n = 37$), astrocytes ($n = 24$), and OPCs ($n = 36$) in the scRNA-seq meta-data. One-way ANOVA, $p = 7.1e{-6}$, ***: $p < 0.001$, ns: not significantly different. Post hoc Tukey HSD comparisons: NSC–Astrocyte, $p = 0.059$; OPC–Astrocyte, $p = 0.00067$; OPC–NSC, $p < 1e^{-7}$ Centre line = median; box = 25th–75th percentiles; whiskers = min–max; outliers shown as points.

NDDs remain largely unexplored. Our study identified that a substantial portion of core matrisome genes (17.2%) and matrisome-associated genes (9.8%) are NDD risk genes, emphasizing the ECM's critical role in cortical development (Fig. 1b). While most of NDD risk matrisome genes are disease-specific, some, such as *LAMA1, LAMA2, RELN, COL4A1, SEMA5,* and *FGF13* are associated with multiple NDDs (Fig. 1d). Future studies on the matrisome associated with NDD risk will provide deeper insights into the core genetic drivers that converge on common ECM-related biological pathways and mechanisms, advancing our understanding of NDD pathogenesis. *COL2A1*, a risk gene for ASD and SCZ, and *NTN1*, a risk gene for ID (Fig. 1d), are notably enriched in the VZ and SVZ of the developing human cortex compared to the developing mouse cortex[27,83]. The expanded oSVZ in humans supports progenitor self-renewal, increased radial glial diversity, and extended neurogenesis, hallmarks of human cortical expansion[1]. This enrichment suggests that these genes may play unique roles in human brain development, potentially contributing to the complexity and specialization of the human brain. The differential expression of *COL2A1* and *NTN1* in humans compared to mice may help explain human susceptibility to NDDs, highlighting species-specific aspects of gene regulation and function during critical stages of cortical development such as progenitor expansion, radial glia diversification and neurogenesis.

Our study provides a comprehensive understanding of matrisome gene expression across different cell types and developmental stages, offering deeper insights into the role of the ECM in brain development. We identified distinct matrisome gene expression signatures for each cell type (Fig. 3 and Supplementary Fig. 3), suggesting that cell-type-specific biological processes and functions are, in part, mediated by the matrisome.

In this study, we demonstrated that matrisome gene expression signatures exhibit temporal dynamics, with the highest number of temporally specific genes observed during the early second trimester across various cell types in the developing cortex (Fig. 4 and Supplementary Fig. 4). Our findings align with established research showing that the human cerebral cortex undergoes significant transcriptional dynamics and extensive cell population diversification during the second trimester[34,84]. Importantly, GO analysis revealed that these signatures are associated with key developmental processes, such as morphogenesis, migration, growth, and cell communication. This highlights the distinct and temporally regulated role of ECM components in guiding critical processes, like cellular differentiation, proliferation, and structural organization during cortical expansion[5–8,13–15,85].

Additionally, our CCI analysis highlights that matrisome genes can exert specialized roles in cell communication in a cell type-selective manner. Notably, non-neuronal cell types, such as tRG, astrocytes, and endothelial cells exhibited strong matrisome-mediated communication, emphasizing their potentially crucial roles during cortical development (Fig. 5).

Furthermore, we found that matrisome gene expression signatures undergo dynamic changes during lineage specification (Figs. 6A, 7A). Characterizing cells expressing these genes provides insights into the molecular mechanisms driving cellular diversity in the developing cortex. Notably, *LGALS3* emerged as one of the matrisome signature genes distinctly different from neural lineage cells, marking subpopulations of NSCs. PCA and epigenomic analyses of *LGALS3* promoter regions revealed shared transcriptomic features with astrocytes, indicating a potential correlation between *LGALS3* expression and the astrocytic lineage specification of NSCs (Fig. 6 and Supplementary Fig. 5). In addition, we identified *S100B*, a well-known astrocyte marker as a matrisome marker gene in OPC during macroglial lineage specification. Future functional studies on these matrisome signatures will offer deeper insights into the molecular mechanism underlying cell type specification during cortical development.

Finally, we identified the cell-type-specific and temporally specific expression of matrisome marker genes associated with NDDs (Fig. 8 and Supplementary Fig. 6). Understanding which matrisome genes are active in specific cell types during distinct time windows enables targeted experiments and interventions, allowing researchers to manipulate gene expression or function in relevant contexts. This information is invaluable for developing therapies targeting specific pathways or cell populations, potentially leading to more precise and effective treatments for developmental brain disorders.

The role of the matrisome during human cortical development remains largely unexplored, and our systematic analyses provide valuable initial insights and resources for further experimental work. However, several limitations must be acknowledged. One limitation is our reliance on transcriptomic data, which does not fully capture the complexity of matrisome protein function, especially given the extensive post-translational modifications such as glycosylation and secretion processes that many ECM components undergo. This is particularly important, as glycosylation can significantly influence the structure, stability, and interactions of ECM proteins[86]. To address these gaps, future research should incorporate proteomics and glycomics[87] approaches to provide a more comprehensive understanding of the matrisome. Spatial proteomics would offer critical insights into the localization and functional contexts of ECM components in their natural milieu within the developing brain. Additionally, understanding the spatial distribution of these proteins will be essential for deciphering the specific roles they play in various regions and cell types during cortical development. Integrating these methodologies will advance our knowledge of how ECM modifications contribute to normal brain development and the pathogenesis of neurodevelopmental disorders, potentially paving the way for future therapeutic strategies.

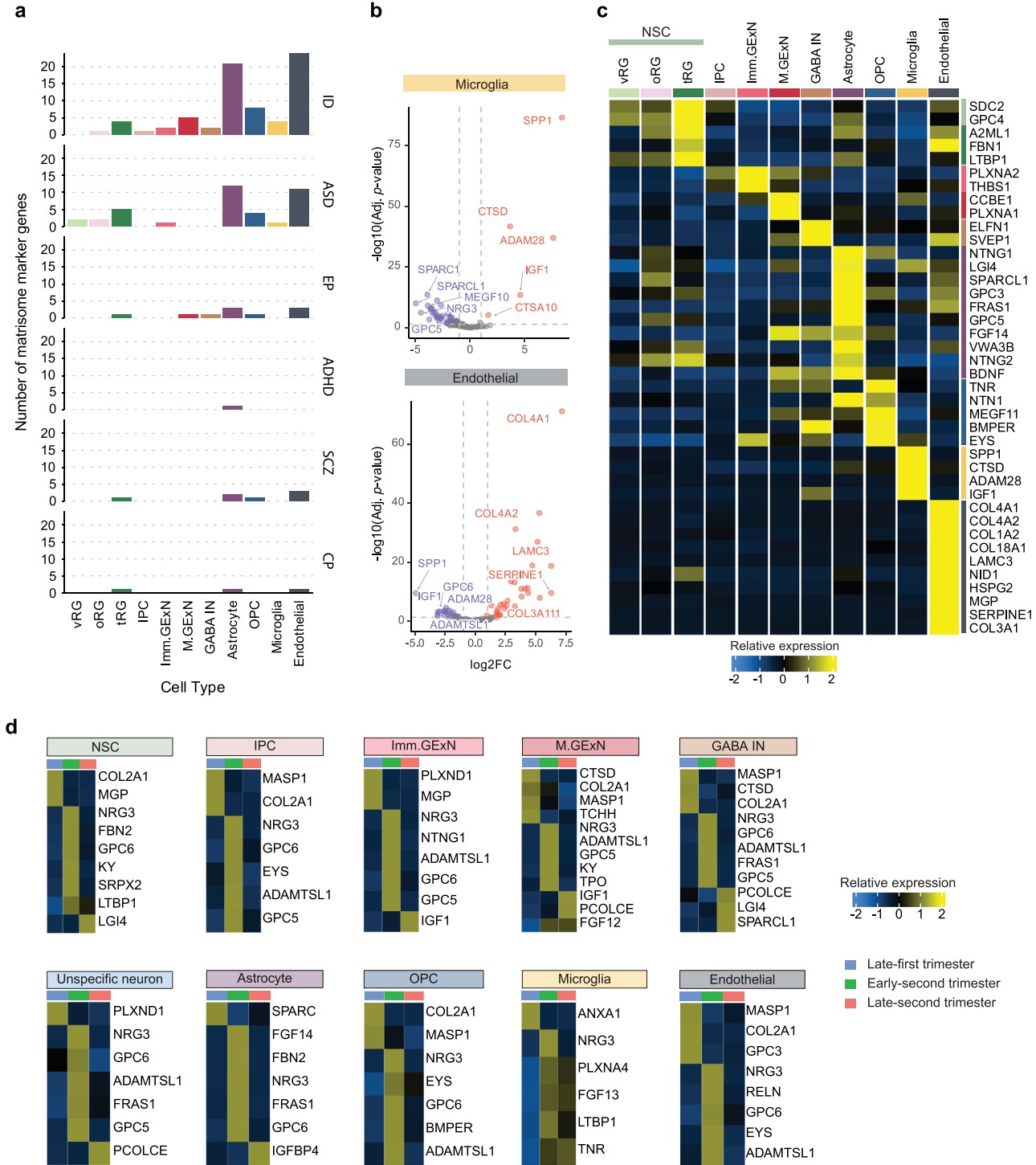

**Fig. 8 | Cell type and temporally specific expression of matrisome genes associated with NDDs. a** Bar graphs displaying number of matrisome marker genes (y-axis) in each cell type (x-axis) associated with NDD. **b** Volcano plot illustrating differentially expressed matrisome genes in microglia and endothelial cells. The top five significantly upregulated and downregulated NDD associated matrisome genes are labeled (ranked by log$_2$ fold-change, adjusted $p$-value < 0.05). **c** Heatmap displaying the expression levels of up to the top five NDD-associated matrisome genes across different cell types (ranked by log$_2$ fold-change, adjusted $p$-value < 0.05). **d** Heatmap displaying the expression levels of up to the top five NDD associated matrisome genes across three developmental periods in all cell type (ranked by log$_2$ fold-change, adjusted $p$-value < 0.05). In **b**–**d**, differential expression was assessed using DESeq2 (negative binomial GLM, two-sided Wald test) with Benjamini–Hochberg FDR correction; effect size is log$_2$ fold-change with 95% Wald confidence intervals ($n$ = as specified in Supplementary Fig. 1d).

## Methods

### Matrisome and NDD risk gene data collection

The Human matrisome database (http://matrisomeproject.mit.edu/other-resources/human-matrisome/)[88] was used to compile a list of 1,027 matrisome genes. Neurodevelopmental risk gene lists were generated by combining data from SFARI Gene (1,162 genes, https://gene.sfari.org/), Geisinger (1183 genes, https://dbd.geisingeradmi.org/), and SysNDD (1,372 gene, https://sysndd.dbmr.unibe.ch/)[37–39].

## scRNA-seq and snRNA-seq data collection

Count matrices from scRNA-seq and snRNA-seq datasets were retrieved from six independent studies[30–35]; Bhaduri et al. and Eze et al. accessed from The Neuroscience Multi-omic (NeMO: https://nemoarchive.org/) data Archive (RRID:SCR_002001 and RRID:SCR_002001), van Bruggen et al. and Cameron et al. from European Genome-phenome Archive (EGA: https://ega-archive.org), under accession numbers EGA: S000 01006136 and EGAS00001006537. The data deposited by Trevino et al. and Zhong et al. were accessed from Gene Expression Omnibus, under accession number, GEO: GSE162170 and GEO: GSE104276. Supplementary Data 13 contains information about number, age, sex, brain region of donors, number of cells in imported matrix, number of cells post quality control, and digital objective identifier (DOI) of each study.

## Single-Cell RNA Sequencing Data Analysis and Integration

Single-cell RNA sequencing (scRNA-seq) data from six independent studies were processed and integrated using the Seurat package v5.1.0[89], tidyverse, dplyr in R v4.4.2[90]. Raw count matrices were filtered to include genes expressed in at least 3 cells and cells with a minimum of 200 detected genes. Quality control was performed on each dataset independently, removing cells with >10% mitochondrial gene content and those with total UMIs or detected genes outside the 2.5th to 97.5th percentile range. The quality-controlled datasets were merged and integrated using Seurat's CCA integration. This involved normalizing each dataset independently using log-transformation with a scale factor of 10,000, identifying 2,000 variable features per dataset by the variance stabilizing transform (vst) method, and selecting highly variable genes across datasets for integration. Integration anchors were computed and used to create an integrated dataset. Post-integration processing included data scaling with regression of RNA counts, principal component analysis (PCA), and construction of a k-shared nearest neighbor (k-SNN) graph using 20 dimensions. Clustering using the Louvain algorithm was performed at a resolution of 1.5, resulting in 40 distinct clusters. UMAP was applied for dimensionality reduction and visualization.

## Cell type annotation

Cell types were annotated using the scType algorithm[40,41], which employs predefined lists of positive and negative marker genes for each cell type. This automated approach assigns cell type labels to clusters based on their gene expression profiles[40]. Marker genes used are shown in Supplementary Data 1.

## Integration quality assessment

To quantitatively assess batch integration and cell type separation, the clustering accuracy with the integration local inverse Simpson's Index (iLISI) and cell-type LISI (cLISI) scores was calculated using a custom function that employs LISI package[42]. The custom function was validated by cross-referencing LISI scores of peripheral blood mononuclear cell (PBMC) data, control (CTRL), and interferon beta-stimulated (STIM)[91], before and after CCA integration described in Stuart and Butler et al., 2018[92].

## Pseudobulk differential gene expression analysis

To mitigate the impact of technical noise and uneven cell sampling across donors during DGE analysis, a pseudobulk matrix was created by aggregating gene expression data for each cell type within each donor, treating one cell type per donor as a single observation. For temporal DGE analysis, a pseudobulk matrix was created by aggregating gene expression data for each trimester per donor.

DGE analysis was performed using DESeq2 v1.44.0 on the pseudobulk data[93]. For each cell type/trimester, a binary condition vector was created, labeling the cell type/trimester of interest as the treatment and all others as the control. A DESeqDataSet object was created using the count matrix and sample meta-dataset. Low-count genes (total counts <10 across all samples) were filtered out. DESeq2 analysis was run using the default parameters. Significantly differentially expressed genes (DEGs) were identified by applying stringent thresholds as an adjusted $p$-value < 0.05 and an absolute log2 fold-change > 2. Results and data generated from this study were visualized using ComplexHeatmap v2.20.0 and ggplot2 v3.5.1[94].

## Gene expression correlation analysis

Correlation analyses between *LGALS3* and *HOPX* gene expression, and between *S100B* and *OLIG2*, were performed using the integrated assay data from the Seurat object, utilizing the FeatureScatter() function to generate scatter plots. Pearson correlation coefficients were calculated to assess the linear relationship between expression levels, with statistical significance evaluated and correlation statistics displayed using the stat_cor() function from the ggpubr package. A linear regression line was fitted using geom_smooth() with method = "lm" to visualise the relationship trend, with each point representing a pseudobulk sample. For the *LGALS3* and *HOPX* correlation analysis, pseudobulk samples comprised NSCs, IPCs, and GExNs, whilst for the *S100B* and *OLIG2* correlation analysis, samples comprised NSCs, astrocytes, and OPCs.

## Gene Ontology (GO) term analysis

Over-representation of gene set analysis was conducted by exporting DGE analysis data to Toppgene (https://toppgene.cchmc.org/)[95]. All detected genes for general DGE analysis and 1,027 matrisome genes for matrisome DGE analysis were used as reference background genes for this analysis. The Enrichment Ratio was computed as the proportion of input genes associated with a specific GO term divided by the total number of input genes. The Annotation enrichment was calculated as the percentage of input genes associated with a GO term to the total number of genes associated with that term in the entire annotation database.

## Weighted gene co-expression network analysis

Weighted Gene Co-expression Network Analysis (WGCNA) was employed to identify modules of co-expressed genes and investigate the network properties of our gene of interest (GOI). Using the WGCNA package v1.73[96], quality control on the expression data was performed, removing genes with excessive missing values or zero variance. The optimal soft-thresholding power was determined by testing a range of powers (1-20) and selecting the lowest power that achieved a scale-free topology fit index ($R^2$2). Using this power value, a signed network was constructed with the blockwiseModules function, setting the minimum module size to 10-30 genes and using a height cut of 0.25 for merging similar modules.

To visualize the network structure around the GOI, the topological overlap matrix (TOM) was calculated using TOMsimilarityFromExpr, and the GOI's module was subset and converted to an igraph object[97]. Based on TOM values, the top genes most strongly connected to the GOI were identified. The resulting network was plotted with nodes representing genes and edge widths proportional to connection strengths using a custom function.

Pearson correlation coefficient and $p$-value between the gene of interest (GOI) and co-expressed gene expression levels were calculated using linear regression with ggpubr v0.6.0.

## Pseudotime analysis

Root cells were set as vRG, and pseudotime for branching cells was calculated based on their distance from the root cells on the principal graph using the monocle3 package v1.3.7 with default parameters[98].

## ChIP-seq and ATAC-seq analysis

To investigate peaks for histone marks and chromatin accessibility, the histone ChIP-seq and ATAC-seq database was accessed from ChIP-

Atlas (https://chip-atlas.org/)[70–72]. The combined peak data was visualized using IGV v2.13.1 software.

### Integrated regulatory network analysis

To identify the regulatory network of our gene of interest, the IReNA (Inference of Regulatory Networks using pseudotime-ordered single-cell RNA-seq data) workflow was employed[75]. The motif data were obtained from the TRANSFAC database[99], and motif-binding transcription factors were extracted. Pseudotime-ordered expression profiles were obtained, filtered for noise, and K-means clustering was performed to group genes with similar expression patterns over pseudotime. Regulatory relationships within grouped genes were identified using Pearson correlation, setting a threshold of $|r| > 0.6$. To enhance the specificity of our predictions, motifmatchr v1.28.0 was used to detect potential TF binding sites in the TSS regions of candidate genes. This analysis was performed using the human genome (GRCh38) as a reference. The regulatory relationships of transcription factors and target genes were filtered based on both correlation and motif binding evidence.

### Temporal dynamics of gene expression analysis

To identify genes with significant temporal expression patterns, a custom R function that analyzes gene expression dynamics across developmental stages was developed. The function performs pseudobulk aggregation of single-cell data by donor and timepoint for each cell type, followed by calculation of Pearson's correlation coefficients and linear regression slopes to quantify expression trends. Genes were filtered for statistical significance ($p < 0.05$) and strong correlation ($|r| > 0.5$), with the top genes selected based on their slope magnitudes.

### Cell-to-cell interaction and pathway analysis

To investigate intercellular communication networks, the CellChat package (v1.6.1) was used[61]. A CellChat object was initialized using the matrisome expression matrix and associated metadata, with cell types as the grouping variable. We utilized the Human CellChatDB as the ligand-receptor interaction database. The dataset was subset to include only genes present in CellChatDB. Over-expressed genes and interactions were identified using CellChat's built-in permutation functions. The communication probability was computed using the trimmed mean approach, and interactions were filtered to include only those involving at least 10 cells. Pathway-level communication probabilities were then calculated.

### Human fetal brain tissue collection

Fetal tissues from elective, normally progressing pregnancies are collected under the Scottish Advanced Fetal Research (SAFeR) study (NCT04613583)[100]. The collection process is approved by one of the 12 Scottish National Health Service Research Ethics Committees (REC 15/NS/0123) and follows the Declaration of Helsinki guidelines. Women seeking elective medical termination of pregnancy were recruited with written informed consent by NHS Grampian research nurses, who operate independently from the research team. There was no alteration in patient treatment or care, and participants could withdraw from the study at any time. The study includes only normally progressing pregnancies, as determined by ultrasound, from women over 16 years of age who speak English and covers gestational ages from 7 to 20 weeks. Grossly abnormal fetuses were excluded, and women experiencing significant distress were not approached. Termination was carried out using RU-486 (Mifepristone) and prostaglandin-induced delivery[101]. Gestational age was confirmed by ultrasound and foot length measurement[102], and various maternal and fetal data were recorded. For sample collection, fetuses are transported to the laboratory within 30 minutes of delivery, typically intact, and are weighed, measured for crown-rump length (CRL), and sexed by morphology and PCR confirmation[70] of the Y chromosome. Collected brain tissues were fixed in 4% PFA in phosphate-buffered overnight at 4 °C followed by preservation in 15% sucrose in PBS at 4 °C overnight, followed by 30% sucrose-sodium azide solution (7.7 mM NaN$_3$). Nine samples were used in this study, including GW 10 (male), GW 11 (male), GW 12 (female), GW 14 (male), GW 15 (female), two GW 16 (both female), and two GW 17 (both male).

### Immunofluorescence staining

The fixed prefrontal cortical samples were embedded in OCT compound and frozen at −80 °C, prior to being cryosectioned at a 20 μm thickness (Leica 1850 croystats). Sections were retrieved onto Superfrost glass slides and stored at −20 °C. Sections were washed in tris-buffered saline (TBS, 137 mM NaCl, 25 mM Tris-HCl and 2.7 mM KCl), and antigen retrieval was performed using Target retrieval solution (Dako, S1699). Sections were permeabilized with 0.05% Triton-X-100 in TBS for 15 minutes at RT and blocked with blocking buffer at room temperature for 1 hour. Primary antibodies were diluted in blocking buffer and incubated with sections over two nights at 4 °C. Sections were washed three times with 0.05% Triton X-100 in TBS and incubated with secondary antibodies diluted in blocking buffer at 4 °C for 2 hours. Sections were washed three times and allowed to dry before treating with Vector TrueView Autofluorescence Quenching Kit with DAPI (Vector, SP-8500), and coverslips were mounted. Antibodies used in this study are listed in Supplementary Data 14.

### Imaging and image analysis

A Zeiss LSM880 confocal microscope and Airyscan Fast with 20X objective were used to acquire images. Z-stacks were collected at 1 μm intervals and image tiles were automatically processed by the inbuilt Zeiss ZEN 3.0 software. Object-based colocalization analysis and semi-supervised cell count were performed using Colocalization Image Creator and Colocalization Cell Counter plugins on ImageJ[103]. The total number of DAPI$^+$ cells was determined using an automatic object counter (radius at 5 microns and noise tolerance at 70), followed by visual inspection of cell counts. The number of HOPX$^+$, S100β$^+$ and OLIG2$^+$ cells was determined by generating a colocalization binary image with DAPI. For Galectin-3$^+$ cell quantification, the grayscale image of Galectin-3 immunoreactivity was merged separately with the DAPI$^+$ binary image and the DAPI-HOPX colocalization binary image. This enabled the identification and quantification of Galectin-3$^+$ cells, as well as Galectin-3$^+$/HOPX$^+$ double-positive cells. The statistical analysis of mean count differences was performed on three biological replicates for HOPX/Galectin-3 staining and four biological replicates for OLIG2 and S100β staining across GW 15-17.

### Reporting summary

Further information on research design is available in the Nature Portfolio Reporting Summary linked to this article.

## Data availability

The raw sequencing data analyzed in this study are available from the original publications in the public repositories indicated therein. Source data are provided as Source_data.xlsx with this paper. Source data are provided with this paper.

## Code availability

All code used in this study is archived and available at Zenodo: https://doi.org/10.5281/zenodo.16811863, which corresponds to the GitHub repository at https://github.com/KANGBERGLAB/matrisome-meta-analysis[104].

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

## Acknowledgements

We thank members of the Kang and Berg laboratories for comments and suggestions. We thank the members of Fowler Lab for support in fetal sample acquisition. This work was supported by grants from The Humane Research Trust Les Rhoades PhD scholarship (to E.K.), The Academy of Medical Sciences Springboard (SBF007\100169 to E.K.), Medical Research Council (MR/Z506138/1 to E.K.), and Biotechnology and Biological Sciences Research Council (BB/W008068/1 to D.A.B.). The SAFeR study was funded by the UK Medical Research Council (MR/L010011/1 and MR/P011535/1) and the EU's Horizon 2020 research and innovation program under the Marie Skłodowska-Curie project PROTECTED (grant agreement number 722634) and FREIA and INITIALISE projects (grant agreement numbers 825100 and 10109409,9 respectively) to PAF.

## Author contributions

E.K., D.A.B., and D.H.G. conceived the project and wrote the paper. D.H.G. led and contributed to all aspects of the study. M.Z.K.A. and M.D.M. provided guidance and suggestions on data analysis. O.S. contributed to immunofluorescence staining of human fetal tissue. P.A.F. contributed to obtaining fetal brain samples. All authors commented on and approved the publication of the manuscript.

## Competing interests

The authors declare no competing interests.
