## [Transparent Peer Review file · Nature Communications]

Deciphering Cell-Type and Temporal-Specific Matrisome Expression Signatures in Human Cortical Development and Neurodevelopmental Disorders via scRNA-Seq Meta-Analysis

Corresponding Author: Dr Eunchai Kang

Version 0:

Reviewer comments:

Reviewer #1

(Remarks to the Author)

This manuscript presents a thorough and comprehensive meta analysis of a number of previously published scRNAseq datasets from developing human forebrain. It provides a detailed catalogue of gene expression that will form a useful resource for future research on how matrisome influences cortical development and likely also highlight possible mechanisms underlying neurodevelopmental disorders.

The data is very clearly presented and explained. As it stands, the work doesn't provide much by way of novel biological insight, its value will most likely lie in providing detailed descriptions of cell-type specific gene expression patterns that will give rise to future hypotheses. Overall, I would describe this as a very solid piece of work that is likely to be of value to the community.

Links to codes used in the extensive bioinformatic analysis are provided, but it is beyond my expertise to validate them.

In the section "Characterisation of neuronal lineage specific changes in matrisome gene expression", the authors describe the developmental positioning of LGALS3+ cells with high HOPX by performing pseudotime analysis using vRG as the root cell type and found that the LGALS3+ cells was at a more mature differentiation state compared to vRG, oRG, tRG and IPC. While it is known that RG cells are the main progenitor cells in the cerebral cortex, it would be interesting to use an independent trajectory inference technique that does not require user input for the root cell such as RNA Velocity to verify their finding.

Minor points:

Can the authors provide some references of the list of marker genes used to annotate the cell types? In particular, there were only 8 genes used to annotate NEC.

Fig 3B, label on x-axis (and in figure legend) says 'unique', but 'distinct' (as used in accompanying text on p9) would be more accurate.

(Remarks on code availability)

Reviewer #2

(Remarks to the Author)

Gim et al have performed a meta-analysis of single cell RNA sequencing (scRNAseq) data from 6 published studies using human fetal cortex or brain. This includes 37 samples of fetal tissue aged 8-26 gestation weeks (GW). The authors have

interrogated this scRNAseq data for matrisome expression, the genes that comprise the core extracellular matrix (ECM) and the ECM-associated proteins, to understand if this changes over cortical development and if the matrisome might be important in neurodevelopmental disorders (NDDs).

This is an interesting question and the ECM has been shown to play many important and instructive roles in cortical development in many different species, including in the human cortex. However, there are major concerns with the current presentation, interpretation and novelty of the results that need to be addressed.

Rationale and language:

1. Throughout the paper many of the statements made lack sufficient detail for the reader to understand the rationale behind the analysis performed or interpretation of the results. Some examples are provided below (not an exhaustive list).

'Notably, a substantial proportion of matrisome genes are associated with NDDs, exhibiting cell type, temporal and disease specificity.'

How many of these matrisome genes are expressed in human fetal cortex? Please clarify exactly which matrisome genes are being referred to in this statement. How do they define cell type, temporal and disease specificity? Three NDD databases were analysed, what is the cut-off for a matrisome gene to be considered to have cell type, temporal and disease specificity? Is this consistent across the three NDD databases? Which matrisome genes are linked to which NDDs?

'This reflects robust matrisome activity during this stage, aligning with the diverse cellular processes occurring at this period.'

What cellular processes are being referred to? Citations should be included to support specific details.

'Collectively, our analyses identified unique, cell-type-specific matrisome signatures, suggesting the potential distinct roles of matrisome components in supporting each cell type's specialized functions during cortical development.'

Further detail is needed on these functions or how these matrisome genes contribute to them.

'This reflects robust matrisome activity during this stage, aligning with the diverse cellular processes occurring at this period.'

Further detail is needed about what cellular processes or matrisome activity the authors are referring to.

Without sufficient detail it is hard to evaluate the novelty and quality of the data and interpretations. This should be improved throughout the manuscript to make it clear what the rationale behind each analysis was and how the authors have developed their interpretation of the results. This should be appropriately cited, giving acknowledgement to findings already published about the ECM in cortical development.

2. Very little detail is provided about the ECM and ECM-associated proteins. Entire families of ECM are often referred to, such as collagens and laminins, when individual sub-chains for each should be described. For example:

'Mutations in ECM-related genes, including POMT1/2 and laminin subunits, are associated with conditions such as cobblestone lissencephaly and polymicrogyria, leading to cortical layer disorganization and abnormal cortical folding''.

Which laminin subunits? This must be detailed and then cited correctly. Are these NDDs included in the NDD databases? Are there other examples of mutations in matrisome genes that were in the NDD databases?

'Furthermore, mutations in ECM components like collagen are linked to malformations such as porencephaly'

Same comment as above.

'The human fetal cortex ECM is more abundant and diverse, especially rich in components such as hyaluronan, chondroitin sulfate proteoglycans, and other glycosaminoglycans, whereas the mouse ECM is less complex²⁴.'

What does less complex mean? More detail should again be provided, and there are more papers available that should be cited to support this claim.

ECM components and associated proteins are also listed without any explanation of what they are, what their function is or what is already known about them. This would be difficult to follow if the reader knew the matrisome field, but given the broad readership of Nature Communications this should be clearly explained to the reader to provide sufficient context. This is critical for some ECM components, where multiple genes need to be expressed to form a functional protein, such as collagen or laminin. Therefore a change in expression of just one sub-chain may not lead to any protein.

3. Some statements are too strong or lack sufficient data to support them. For example:

'This technology facilitates the analysis of cellular interactions across various developmental stages, providing detailed insights into the progression of cellular differentiation and development'

Please clarify what detailed insights are gained. Without lineage tracing, does this meta-analysis provide such detailed descriptions of the progression of differentiation and development, or does it suggest them?

‘Additionally, our CCI analysis highlights that matrisome genes can exert specialized roles in cell communication in a cell type-selective manner’

This refers to astrocytes and truncated radial glia, which appear later in development. How can the authors rule out other interactions that may occur prior to the generation of astrocytes? Please clarify what specialised roles are being referred to and add citations where appropriate.

‘Furthermore, we found that matrisome gene expression signatures undergo dynamic changes during lineage specification’

‘LGALS3 emerged as one of the matrisome signature genes distinctly different from neural lineage cells, marking subpopulations of NSCs’

These statements refer to LGALS3, which did not show lineage specific expression in the data presented, but was instead found in both basal radial glia and astrocytes (and shown in the marginal zone by immunofluorescence in Fig 6). Please clarify these points or change these statements.

‘COL2A1, a risk gene for ASD and SCZ, and NTN1, a risk gene for ID (Fig. 1d), are notably enriched in the VZ and SVZ of the developing human cortex compared to the developing mouse cortex^{25,75}. This enrichment suggests that these genes play unique roles in human brain development, potentially contributing to the complexity and specialization of the human brain.’

Please clarify how this enrichment suggests this. This statement should be made more specific and include appropriate citations to support it.

‘The differential expression of these genes in humans compared to mice may help explain human the susceptibility to NDDs, highlighting species-specific aspects of gene regulation and function during critical stages of cortical development.’

No citations have been provided. This statement should be made more specific and include appropriate citations to support it. How are they differentially expressed? What critical stages of cortical development?

‘Our study provides a comprehensive understanding of matrisome gene expression across different cell types and developmental stages, offering deeper insights into the role of the ECM in brain development.’

This statement should be made more specific and include appropriate citations to support it.

Data analysis:

4. No supplementary data has been provided for the quality control the authors performed on the publicly available datasets. How many samples were taken or left from each of these datasets? How many cells were excluded? Has any analysis been performed to account for the weighting of the number of cells within an individual sample or study? For example, there are 213,659 cells analysed, yet around 50,000 of those appear to be from one 20 GW sample (Fig 2b). What has been done to ensure this does not bias the results? If there is a bias, could further datasets be added?

Fig S1d shows donors per cell type, but it would be useful to see the ages of each sample contributing to each cell type, and how this is proportionally shared between samples. If the majority of a specific type of progenitor are only found in one sample, or one age, this could lead to bias in later analysis, i.e. very few ventral radial glial are found in samples around 26 GW, and those that are present will behave very differently to ventral radial glia found at 9 GW. This links to the analysis of the heterogeneity of NSC subtypes seen in Fig. S2b.

5. The exact regions and tissue types used for each dataset should be stated and clearly discussed. Has just cortical tissue been used or whole brain? Which area of the cortex? Is this different between different aged samples? Could any differences in gene expression identified be linked to cortical regions and not age?

6. The rationale for the age ranges selected for the temporal analysis is not clear. Have these been selected based on biological processes? How many samples and cells are in each age range? How many NSCs are in each age range? How many samples contribute to each cell type/group within these age ranges? Could differences in this explain some of the gene expression changes identified? I.e. if the predominant NPC subtype has changed with age, this could explain the changes in the NSC described.

7. It is not fully clear why some of the highlighted genes have been selected, with some previously published yet not cited well. TNC has already been shown to be a marker of basal radial glia (Pollen et al, 2024) and the LGALS3 related gene LGALS3BP has been linked to cortical expansion (Kyrousi et al, 2021).

‘While S100B is known to be expressed in OPCs and immature oligodendrocytes in the developing mouse brain^{68,69}, its expression dynamics in OPCs during human cortical development remain poorly understood.’

S100b has been studied in human brain development (reviewed by Hernández-Ortega et al, 2024), and has been described to be expressed in human OPCs and oligodendrocytes.

8. The immunofluorescent images provided can be improved. In Fig 6, many cells appear to express LGALS3 but not Hopx (Fig 6d), including a strong band of staining in the marginal zone. Can the authors provide more detail on this? Why were the ages of tissue used selected? Does this align with the expression from the scRNAseq? The exact expression levels for LGALS3 and Hopx in these datasets should be included. The image quality of Fig.6g is very low and it is hard to see the expression clearly, however LGALS3 appears to be in the intermediate zone at 14 GW, not the progenitor zones. These images should include a progenitor marker to evaluate this, such as Sox2 or Hopx.

9. Further validation for cell-type specific matrisome expression and temporal changes are needed to support the statements made. This could be via immunofluorescence or ISH for the top candidates. In Fig. 4b, as the major changes described occur during the early second trimester, it would be interesting to see how expression levels of each matrisome component highlighted change in each age examined to understand the dynamics of this matrisome increase. In Fig. 4d, the GO analysis for general terms could also be complemented with analysis of specific ECM functions (similar to that shown in Fig 1a).

10. 'Non-neuronal lineage cell types exhibit the strongest tendency toward positive temporal regulation, with endothelial cells showing an 85.0% (17 out of 20 genes) and OPCs displaying 80.6% (25 out of 31 genes) of temporally increasing matrisome genes, suggesting their growing contribution to matrisome-mediated functions during cortical development'

The authors do not comment on the type of matrisome genes expressed by endothelial cells. These cells produce a vast amount of vascular basement membrane as blood vessels develop, and it is not surprising that this increases over the course of development, as blood vessels grow and mature and the basement matures with them.

11. No validation has been provided for the ligand-receptor interactions proposed from the CCI analysis. Immunofluorescence data are needed to support these statements, along with any citations available to show that these pathways are active in the cell types listed.

12. 'Astrocytes, M.GExN, and NSCs had the highest number of temporal marker genes, suggesting significant contribution of matrisome gene expression during development.'

Is this due to fact that the numbers/proportions of these cells change greatly over 8-26gw?

13. Is the pseudotime analysis performed based solely on the authors own clustering? Temporal information is available for each sample analysed, this should also be used for an in depth analysis of gene expression changes over time for LGALS3.

Minor comments:

1. The figure legend from Fig. S2 is incomplete (missing 2c-e).
2. Details of the statistical tests in Fig. 6e, 7f are missing (t-tests?)
3. Fig. 6d lacks a box to show where 'd' is located.
4. The SVZ in human is compared to the VZ in the mouse, what age is this? Is this the OSVZ or ISVZ?
5. Few papers are cited in the introduction for the rationale of examining the ECM in the human fetal cortex. More should be included to represent the field and volume of work. For example, what ECM in the VZ is known to drive proliferation?

(Remarks on code availability)

Reviewer #3

(Remarks to the Author)

The ECM is an essential component for multi-cellular organisms. In this manuscript, Gim et al. utilized a meta-analysis of scRNA-seq datasets of human cortical development to systemly characterize gene expression signatures related to matrisome. With this analysis, the authors found matrisome gene expression showing both cell-type and developmental stage-specific features, highlighting the important role of ECM in cortical development and diseases. Overall, the concept of this work is meaningful, and the analysis is rigorous. Below are some suggestions for the authors before the manuscript is considered for publication.

1. Since the overlapping of matrisome genes and NDD risk genes is a major delivery, Fig. 1b should perform a significant test.

2. The authors extensively discuss the existence of cell-type or temporal-specific features of matrisome genes. However, the underlying gene regulation mechanisms were not discussed. Fig. 6k identified potential TFs driving LGALS3 expression. Since all the data come from scRNA-seq, it will be worth to check whether there is a co-expression significance between LGALS3 and its potential TFs. The same analysis may apply to other matrisome genes to imply an upstream gene regulation.

(Remarks on code availability)

Reviewer #4

(Remarks to the Author)

(Remarks on code availability)

Reviewer #5

(Remarks to the Author)

(Remarks on code availability)

Version 1:

Reviewer comments:

Reviewer #1

(Remarks to the Author)

As before, the manuscript presents a thorough and comprehensive meta analysis of a number of previously published scRNAseq datasets from developing human forebrain. It provides a detailed catalogue of gene expression that will form a useful resource for future research on how matrisome influences cortical development and likely also highlight possible mechanisms underlying neurodevelopmental disorders.

The data is clearly presented and explained. As it stands, the work doesn't provide much by way of novel biological insight, its value will most likely lie in providing detailed descriptions of cell-type specific gene expression patterns that will give rise to future hypotheses. Overall, I would describe this as a very solid piece of work that is likely to be of value to the community.

I confirm that the authors have satisfactorily addressed the minor concerns I raised on the previous version, I would be happy to see the manuscript published without further changes.

(Remarks on code availability)

Reviewer #2

(Remarks to the Author)

Please see uploaded file with colour coded responses for clarity.

(Remarks on code availability)

Reviewer #3

(Remarks to the Author)

The authors have addressed my concerns. The manuscript has been significantly improved and is now suitable for publication.

(Remarks on code availability)

Reviewer #4

(Remarks to the Author)

(Remarks on code availability)

Reviewer #5

(Remarks to the Author)

(Remarks on code availability)

cortex. However, there are major concerns with the current presentation, interpretation and novelty of the results that need to be addressed.

We thank the reviewer for the detailed summary of our study and for recognizing the relevance of our investigation into matrisome expression during human cortical development. We agree that the extracellular matrix plays a critical and multifaceted role in neurodevelopment, and we appreciate the reviewer's acknowledgment of the significance of this research question.

While we acknowledge the reviewer's concerns regarding the presentation, interpretation, and novelty of the results, we believe that many of these points stem from areas where further clarification was needed. Accordingly, we have revised the manuscript to improve clarity, better articulate the context and significance of our findings, and address the reviewer's specific comments, as outlined below.

Rationale and language:

1. Throughout the paper many of the statements made lack sufficient detail for the reader to understand the rationale behind the analysis performed or interpretation of the results. Some examples are provided below (not an exhaustive list).

'Notably, a substantial proportion of matrisome genes are associated with NDDs, exhibiting cell type, temporal and disease specificity.'

How many of these matrisome genes are expressed in human fetal cortex? Please clarify exactly which matrisome genes are being referred to in this statement. How do they define cell type, temporal and disease specificity? Three NDD databases were analysed, what is the cut-off for a matrisome gene to be considered to have cell type, temporal and disease specificity? Is this consistent across the three NDD databases? Which matrisome genes are linked to which NDDs?

We thank the reviewer for their detailed comments and questions regarding the specificity and interpretation of matrisome gene expression in relation to NDDs.

The sentence the reviewer referring to is in the abstract and all the detailed information is described in the main text, methods and figure legends in our original manuscript.

Our integrated scRNA-seq dataset, comprising 37 human fetal cortical samples from six independent studies spanning GW8 to GW26, includes expression of 953 out of 1,027 human matrisome genes (**as described in main text, Supplementary Fig. 3 and Supplementary Fig. 3 legend**). We acknowledge the distinction between undetected and truly unexpressed genes and have taken care to include only reliably detected genes in our downstream analyses.

Throughout the manuscript, we identify cell type, temporal and disease specific matrisome marker genes using differential gene expression analysis, as described in the Methods section and figure legend. Marker genes are defined based on a \log_2 fold-change threshold and an adjusted p-value < 0.05 , and their specificity is illustrated in heatmaps (**Fig. 3c, Supplementary Table 3, Fig. S4, Supplementary Table 6, and Fig. 4c**).

To identify matrisome genes associated with NDDs, we intersected the curated human matrisome gene set (1,027 genes) with a combined list of 2,723 unique NDD risk genes compiled from three publicly available NDD gene databases. This intersection yielded the subset of matrisome genes linked to NDDs. We illustrate these associations using a chord plot (**Fig. 1d**), and further detail which matrisome marker genes, defined by cell type or

temporal specificity, are associated with specific NDDs in **Fig. 8** and **Supplementary Tables 10 to 12**.

'This reflects robust matrisome activity during this stage, aligning with the diverse cellular processes occurring at this period.'

What cellular processes are being referred to? Citations should be included to support specific details.

We thank the reviewer for this helpful comment. To clarify, the cellular processes referred to during the early second trimester (GW13 to GW19) developmental window include active neurogenesis, radial glial scaffold maturation, emergence of oRG and tRG neuronal migration, and the early stages of gliogenesis and cortical lamination. These processes are closely associated with dynamic extracellular matrix remodeling and support the observed increase in matrisome activity. In response to the reviewer's suggestion, we have revised the manuscript to specify these cellular events and have included relevant citations to support this statement.

'Collectively, our analyses identified unique, cell-type-specific matrisome signatures, suggesting the potential distinct roles of matrisome components in supporting each cell type's specialized functions during cortical development.'

Further detail is needed on these functions or how these matrisome genes contribute to them.

We thank the reviewer for this comment. The sentence in reviewer's comment is intended as a summary statement, with detailed descriptions of the specific functions and gene-level contributions presented in the preceding results and associated figures. For example, in **Fig. 3c–d**, we highlight distinct matrisome marker genes for each cell type, such as *RELN* and *COL1A2* in vRG (associated with neuronal migration and ECM remodeling), *SPARCL1* and *FBLN2* in oRG (involved in synaptic organization and neurogenesis), and *NRG3* and *ITIH2* in tRG (linked to progenitor signaling and ECM integrity). These findings are supported by relevant functional annotations and literature references in the manuscript. While we have highlighted representative examples, a comprehensive functional interpretation of all matrisome gene signatures across cell types is beyond the scope of the current study. However, we agree that this is an important direction for future work.

Without sufficient detail it is hard to evaluate the novelty and quality of the data and interpretations. This should be improved throughout the manuscript to make it clear what the rationale behind each analysis was and how the authors have developed their interpretation of the results. This should be appropriately cited, giving acknowledgement to findings already published about the ECM in cortical development.

We thank the reviewer for this feedback. We fully agree that clearly articulating the rationale behind each analysis and how our interpretations were developed is essential for evaluating the novelty and quality of our findings. In response, we have carefully revised the manuscript to improve the clarity and coherence of our analytical framework and explicitly state the rationale for each major analysis step. We have also included additional citations to

appropriately acknowledge prior studies on the role of the ECM in cortical development. These revisions aim to clarify how our work builds upon existing knowledge and highlight the unique insights offered by our integrative meta-analysis approach.

2. Very little detail is provided about the ECM and ECM-associated proteins. Entire families of ECM are often referred to, such as collagens and laminins, when individual sub-chains for each should be described. For example:

‘Mutations in ECM-related genes, including POMT1/2 and laminin subunits, are associated with conditions such as cobblestone lissencephaly and polymicrogyria, leading to cortical layer disorganization and abnormal cortical folding”.’

Which laminin subunits? This must be detailed and then cited correctly. Are these NDDs included in the NDD databases? Are there other examples of mutations in matrisome genes that were in the NDD databases?

‘Furthermore, mutations in ECM components like collagen are linked to malformations such as porencephaly’

Same comment as above.

We thank the reviewer for this constructive comment. We agree that more specificity is warranted when discussing ECM-related proteins, particularly when referring to broad families such as laminins and collagens.

In response, we have revised the manuscript to provide greater detail on the individual ECM subunits and components discussed. Specifically, we now specify the relevant laminin subunits (LAMB2, LAMC3) and collagen genes (COL4A1, COL4A2, COL18A1) that have been implicated in cortical malformations such as cobblestone lissencephaly, polymicrogyria, and porencephaly.

Additionally, we have cross-referenced these specific genes with those included in the three NDD databases analyzed in our study. They are all included in the databases.

‘The human fetal cortex ECM is more abundant and diverse, especially rich in components such as hyaluronan, chondroitin sulfate proteoglycans, and other glycosaminoglycans, whereas the mouse ECM is less complex²⁴.’

What does less complex mean? More detail should again be provided, and there are more papers available that should be cited to support this claim.

We thank the reviewer for raising this important point. In the paper we originally cited¹, the authors state that “the extraordinary size and complexity of the human cerebral cortex are the result of a sophisticated and exquisitely orchestrated developmental program, which emerged during mammalian evolution,” and that “each mammalian species expresses in cortical germinal zones a unique combination of ECM components at unique relative levels.” This is supported by data in their Table 1, which reports 37 ECM genes expressed in the human cortex compared to 27 in the mouse, indicating broader diversity of ECM components in the human context.

In this context, the term "less complex" refers to both the reduced diversity and lower abundance of certain ECM molecules in the mouse fetal cortex compared to human. For

example, the human cortex contains a wider variety and higher levels of molecules such as hyaluronan, chondroitin sulfate proteoglycans, and other glycosaminoglycans. In contrast, the mouse ECM contains fewer types and lower amounts of these components. These differences likely contribute to species specific differences in how the ECM supports cortical development, structural organization, and cellular signaling.

In response to the reviewer's suggestion, we have now cited additional literature to better support this comparison. For example, Pokhilko et al. conducted a global proteomic analysis of brain ECM and reported that the human cerebrovascular ECM includes more collagens (11 versus 10), glycoproteins (36 versus 26), and proteoglycans (6 versus 5) than mouse². Both species showed similar numbers of basement membrane proteins (20 in human and 19 in mouse), but the broader ECM proteome in human tissue demonstrated greater complexity. These findings support and complement the transcriptomic differences observed in earlier studies.

We have revised the relevant section of the manuscript to clarify the intended meaning and included the additional citation to strengthen our argument.

ECM components and associated proteins are also listed without any explanation of what they are, what their function is or what is already known about them. This would be difficult to follow if the reader knew the matrisome field, but given the broad readership of Nature Communications this should be clearly explained to the reader to provide sufficient context. This is critical for some ECM components, where multiple genes need to be expressed to form a functional protein, such as collagen or laminin. Therefore a change in expression of just one sub-chain may not lead to any protein.

We thank the reviewer for raising this important point. We agree that providing sufficient context is essential for a broad readership, particularly given the complexity and diversity of matrisome components and their roles in cortical development.

Matrisome proteins carry out diverse and context dependent functions that vary based on their spatial, temporal, and cell type specific expression. Our study focuses on characterizing the expression patterns of these genes during human cortical development and aims to serve as a foundation for future mechanistic investigations. To this end, we have taken a systemic level, data driven approach to identify reproducible matrisome expression signatures using integrated analyses across six independent studies and 37 fetal cortex donors.

While we have included references to known functions of selected matrisome genes where relevant and supported our findings with GO term analysis, the primary goal of our study is computational hypothesis generation, rather than exhaustive functional annotation. We agree that some ECM components, such as collagens and laminins, require coordinated expression of multiple subunits to form functional proteins. To address this, we now clarify this point in the manuscript and have added relevant caveats when interpreting differential expression of individual subunits.

We also note that the general functions of individual matrisome genes are curated and publicly available through resources such as the Matrisome Project^{3,4}. We have provided full lists of differentially expressed matrisome genes in each analysis, which can serve as a resource to guide future functional validation studies in the context of human brain development.

In response to the reviewer's comment, we have revised the manuscript to improve clarity for readers less familiar with the matrisome field and to emphasize the interpretive boundaries of our approach.

3. Some statements are too strong or lack sufficient data to support them. For example:

'This technology facilitates the analysis of cellular interactions across various developmental stages, providing detailed insights into the progression of cellular differentiation and development'

Please clarify what detailed insights are gained. Without lineage tracing, does this meta-analysis provide such detailed descriptions of the progression of differentiation and development, or does it suggest them?

This statement in the introduction was intended to highlight the capabilities of single cell RNA sequencing and how it has been commonly used in previous studies, such as summarized in the review paper by Vinsland et al.⁵, to investigate cellular interactions and the progression of cellular differentiation and development at single cell resolution. Our aim in including this statement was to provide a rationale for leveraging scRNA-seq data to explore cell type and temporal specific matrisome expression signatures in the developing human cortex, rather than to imply that our meta-analysis alone offers direct lineage-level insight.

'Additionally, our CCI analysis highlights that matrisome genes can exert specialized roles in cell communication in a cell type-selective manner'

This refers to astrocytes and truncated radial glia, which appear later in development. How can the authors rule out other interactions that may occur prior to the generation of astrocytes? Please clarify what specialised roles are being referred to and add citations where appropriate.

We thank the reviewer for this comment. Our cell-cell interaction (CCI) analysis, performed using CellChat, estimates interaction strength based on the expression levels of ligands and receptors across all major cell types in the developing human cortex from GW 8 to 26. This method infers signaling pathways in an unbiased manner, and during statistical testing, certain interactions, particularly those that are temporally restricted or low in expression, may be filtered out.

While we emphasize astrocytes and tRG as the most prominent contributors to matrisome mediated interactions due to their relatively high inferred interaction strengths, our analysis does not exclude interactions involving earlier cell types. As shown in **Fig. 5a and 5d**, other cell populations, including vRG, oRG, IPC, OPC, and neurons, also participate in matrisome associated signaling, though with lower interaction strengths compared to tRG and astrocytes.

The sentence the reviewer mentioned appears in the Discussion section, where we aim to summarize the findings from the meta-analysis. The term "specialized roles" refers to the selective involvement of matrisome ligands and receptors, such as PTN and SDC2, which contribute to extracellular signaling in a cell type specific and developmentally regulated manner as we described in the result section.

'Furthermore, we found that matrisome gene expression signatures undergo dynamic changes during lineage specification'

We thank the reviewer for this question. The sentence the reviewer mentioned appears in the Discussion section. This statement refers to the dynamic and cell fate dependent shifts in matrisome gene expression observed during lineage specification, as illustrated in **Fig. 6a** and **Fig. 7a**. For example, NSCs that differentiate into IPCs or neurons downregulate *TNC* while upregulating genes such as *SMOC1* and *LAMB1* (**Fig. 6a**). Similarly, NSCs initially express *LOX* and *ANXA2*, but these genes are downregulated as the cells commit to astrocyte or OPC fates, which instead express *SPARCL1* and *S100B* (**Fig. 7a**). These observations reflect the coordinated and stage specific modulation of matrisome gene expression as NSCs transition into lineage restricted progenitors and terminal cell types. The examples supporting this statement are detailed in the Results section.

'LGALS3 emerged as one of the matrisome signature genes distinctly different from neural lineage cells, marking subpopulations of NSCs'

These statements refer to LGALS3, which did not show lineage specific expression in the data presented, but was instead found in both basal radial glia and astrocytes (and shown in the marginal zone by immunofluorescence in Fig 6). Please clarify these points or change these statements.

We thank the reviewer for this helpful comment and the opportunity to clarify our interpretation of LGALS3 expression. The sentence the reviewer mentioned appears in the Discussion section, where we aim to summarize the findings from **Fig. 6**. As shown in **Fig. 6a**, LGALS3 is expressed by NSCs and is downregulated in IPCs and neurons, suggesting a distinct expression pattern from the neuronal lineage. We have confirmed this observation through immunofluorescence, which is now included in the revised **Supplementary Fig. 6a**. However, we do not intend to imply that LGALS3 is restricted to a single lineage. Rather, our findings indicate that LGALS3 marks a subpopulation of NSCs and oRGs with enrichment toward a glial lineage trajectory.

Additionally, we observed that not all NSCs express Galectin-3 protein. Quantitative analysis revealed that $71.1 \pm 4.2\%$ of HOPX-positive cells in the GW16–17 prefrontal cortex co-express Galectin-3. This was assessed using an unsupervised, automated method that counted DAPI-positive nuclei co-localized with Galectin-3 and HOPX. Importantly, this approach focused on cell bodies and excluded Galectin-3 localized in glial processes or extracellular matrix components, particularly in regions like the intermediate and marginal zones.

We also acknowledge the reviewer's insightful comment regarding LGALS3 expression in the marginal zone. The marginal zone contains microglia, which are well-documented sources of LGALS3⁶. Additionally, the endfeet of radial glia, particularly oRGs, may contribute to LGALS3 expression observed in this region. Furthermore, we have detected LGALS3 localization within the ECM.

The marginal zone is a specialized, signaling-rich, and ECM-dense environment that serves as a critical interface between the developing cortex and the overlying meninges. Thus, LGALS3 expression may also reflect the presence or secretion of Galectin-3 by other known LGALS3-expressing cell types in or near this interface, including macrophages, myeloid-derived cells, fibroblasts, and meningeal microglia-like cells⁷⁻¹¹. We have revised the figure legend accordingly to clarify this point and prevent potential misinterpretation.

Taken together, these findings support our interpretation that LGALS3 marks a subpopulation of NSCs or oRGs and exhibits a distinct, though not exclusive, expression signature that differentiates it from the neuronal lineage.

'COL2A1, a risk gene for ASD and SCZ, and NTN1, a risk gene for ID (Fig. 1d), are notably enriched in the VZ and SVZ of the developing human cortex compared to the developing mouse cortex^{25,75}. This enrichment suggests that these genes play unique roles in human brain development, potentially contributing to the complexity and specialization of the human brain.'

Please clarify how this enrichment suggests this. This statement should be made more specific and include appropriate citations to support it.

We thank the reviewer for this comment. We agree that the original statement would benefit from additional clarification and supporting references.

COL2A1 and *NTN1* are matrisome genes identified as risk factors for NDDs in all three curated NDD gene databases used in our study. As described in previously cited studies^{12,13}, these genes are enriched in the ventricular and subventricular zones (VZ and SVZ) of the developing human cortex, but not to the same extent in the developing mouse cortex. This enrichment is associated with the expanded outer SVZ in humans, which supports prolonged progenitor self-renewal, increased radial glial diversity, and extended neurogenesis—hallmarks of human cortical expansion.

COL2A1, a collagen gene, and *NTN1*, a guidance cue, contribute to a human-specific ECM microenvironment that facilitates these processes¹⁴. In contrast, the mouse ECM transcriptome in comparable regions shows lower complexity and undergoes an earlier transition to a cortical plate-like profile. Therefore, the human-enriched expression of *COL2A1* and *NTN1* is thought to support expanded cortical neurogenesis and circuit refinement, while pathogenic variants in these genes may disrupt these finely regulated developmental events, predisposing individuals to NDDs.

In response to the reviewer's suggestion, we have revised the manuscript to clarify that this enrichment suggests, rather than confirms, species-specific adaptations in ECM and guidance mechanisms that contribute to human brain specialization. We have also ensured that the relevant citations are included to support this interpretation.

'The differential expression of these genes in humans compared to mice may help explain human the susceptibility to NDDs, highlighting species-specific aspects of gene regulation and function during critical stages of cortical development.'

No citations have been provided. This statement should be made more specific and include appropriate citations to support it. How are they differentially expressed? What critical stages of cortical development?

We thank the reviewer for this comment and appreciate the opportunity to clarify this point. The statement was intended as a concluding remark following the previous paragraph, where we describe the enrichment of *COL2A1* and *NTN1* in the human ventricular and subventricular zones (VZ and SVZ), compared to the developing mouse cortex. This enrichment is supported by transcriptomic comparisons presented in published studies^{12,13}, which highlight differential gene expression in cortical progenitor zones across species.

To address the reviewer's concern, we have revised the statement to make it more specific and to clearly link it to these references.

'Our study provides a comprehensive understanding of matrisome gene expression across different cell types and developmental stages, offering deeper insights into the role of the ECM in brain development.'

This statement should be made more specific and include appropriate citations to support it.

We thank the reviewer for this helpful comment. This statement was intended as a summary of our analytic findings, which characterize matrisome gene expression across multiple cell types and developmental stages in the human fetal cortex. As a concluding sentence, it reflects the overall scope of the study rather than presenting new data. The specific details and supporting evidence for this summary are provided in the Results section, where we describe the cell type-specific signatures (**Fig. 3**), temporal dynamics (**Fig. 4**), and cell-cell interaction patterns (**Fig. 5**) in detail, with citations included where appropriate in the result section and the following paragraphs in the discussion.

Data analysis:

4. No supplementary data has been provided for the quality control the authors performed on the publicly available datasets. How many samples were taken or left from each of these datasets? How many cells were excluded?

We thank the reviewer for this comment. Information regarding quality control, including the number of samples included from each dataset and the number of cells excluded during preprocessing, is provided in **Supplementary Table 13**. This table summarizes dataset-level filtering criteria and cell counts following quality control steps. We have now clarified this reference in the manuscript to ensure it is easier for readers to locate.

Has any analysis been performed to account for the weighting of the number of cells within an individual sample or study? For example, there are 213,659 cells analysed, yet around 50,000 of those appear to be from one 20 GW sample (Fig 2b). What has been done to ensure this does not bias the results? If there is a bias, could further datasets be added?

We thank the reviewer for raising this important point. We agree that disproportionate distribution of cell numbers across samples can potentially bias scRNA-seq analyses, particularly due to issues such as pseudo-replication, where cells from the same donor are treated as independent observations. This is a well-recognized limitation in single cell studies due to the inherent variability in cell yield, resolution, and cell type composition between samples.

To address this concern, we implemented a pseudobulk approach, as shown in **Fig. 3a** and detailed in the Methods section. Specifically, we aggregated gene expression values across all cells of a given cell type within each donor, treating each donor-cell type combination as a single observation. For temporal analyses, we further aggregated gene expression by trimester within each donor. This strategy ensures that each sample contributes equally, regardless of the number of cells it contains, thereby mitigating the potential bias introduced by overrepresented samples such as the 20 GW dataset.

Differential gene expression was then performed on the resulting pseudobulk matrices using DESeq2 (v1.44.0), a well-established method for bulk transcriptomic analysis that effectively

controls sample-level covariates and reduces batch effects. For identifying temporally regulated genes, we developed a custom function that computes Pearson correlation coefficients and linear regression slopes on the pseudobulked data, stratified by developmental stage and cell type. Only genes with $|r| > 0.6$ and an adjusted p-value ≤ 0.05 were considered for downstream interpretation.

This approach not only corrects for uneven cell sampling across studies and donors but also enhances biological interpretability by reducing cell-level noise. We believe that this method effectively addresses concerns related to sample imbalance and that our key findings are not biased by the number of cells per sample.

In light of the reviewer's question, we have clarified this methodological choice more explicitly in the revised manuscript to ensure that the rationale and benefits of the pseudobulk approach are clearly understood.

Fig S1d shows donors per cell type, but it would be useful to see the ages of each sample contributing to each cell type, and how this is proportionally shared between samples. If the majority of a specific type of progenitor are only found in one sample, or one age, this could lead to bias in later analysis, i.e. very few ventral radial glial are found in samples around 26 GW, and those that are present will behave very differently to ventral radial glia found at 9 GW. This links to the analysis of the heterogeneity of NSC subtypes seen in Fig. S2b.

We thank the reviewer for this comment. We agree that unequal representation of specific progenitor types across developmental stages could introduce bias if individual cells were treated as independent observations. However, to avoid this issue, we employed a pseudobulk approach, in which gene expression data were aggregated for each cell type within each donor. This means that, irrespective of how many cells were present in each sample, all donor–cell type combinations were treated as single observations in the differential gene expression analysis. This method ensures that each donor contributes equally to the analysis and minimizes the influence of uneven cell numbers across developmental stages.

As shown in **Supplementary Fig. 1d**, each cell type is contributed by multiple donors spanning different gestational ages, reflecting a relatively proportional distribution across developmental time. This mitigates the risk that any particular cell type is disproportionately represented by a single donor or developmental stage.

A similar pseudobulk strategy was applied to the differential expression analysis of NSC subtypes shown in **Supplementary Fig. 2b**. Therefore, we interpret the heterogeneity observed among NSC subtypes as biologically meaningful rather than an artifact of sampling imbalance. Additionally, UMAP clustering was performed using CCA for data integration, and this was validated using the Local Inverse Simpson's Index in **Fig. 2c** and **Fig. 2d**, further reducing potential donor-specific or stage-specific biases.

To ensure robustness in our temporal analyses, we also grouped gestational ages into three broad developmental windows—late first trimester, early second trimester, and late second trimester—rather than analyzing data at individual gestational weeks. This strategy was chosen to reduce noise from age-to-age variability while ensuring sufficient cell type representation across each developmental period.

We have now clarified these points in the manuscript to better explain how our approach addresses the reviewer's concern regarding sampling imbalance across developmental stages.

5. The exact regions and tissue types used for each dataset should be stated and clearly discussed. Has just cortical tissue been used or whole brain? Which area of the cortex? Is this different between different aged samples? Could any differences in gene expression identified be linked to cortical regions and not age?

We thank the reviewer for this important comment. We agree that brain region is a key consideration in cell type specific transcriptomic analyses, particularly when interpreting developmental and spatial expression differences.

In our study, we primarily selected datasets derived from the cortex, which was the most consistently available region across studies. However, one of the datasets (van Bruggen et al., 2022) includes transcriptomic data from the forebrain. For greater clarity, we have now included detailed regional information for each sample in **Supplementary Table 13**.

Importantly, even though one dataset includes a broader cortical region, each cell type and developmental stage is independently replicated across multiple datasets, helping to minimize potential bias from regional variation. Moreover, our use of pseudobulk gene expression matrices aggregates expression at the donor level and reduces the influence of local regional heterogeneity within individual samples. Differential gene expression was then assessed using DESeq2, which accounts for variability across donors and covariates between studies.

We further note that any variation in gene expression due to cortical region rather than developmental stage or cell type would result in greater intra-group variance and therefore higher adjusted p-values. As a result, only genes with consistent patterns across donors and samples—reflected in an adjusted p-value ≤ 0.05 and $|r| > 0.6$ for temporal trends—were considered in downstream analyses.

Based on these methodological steps, we believe that any regional variability present in a subset of the datasets is unlikely to have significantly influenced the overall analytic outcomes.

6. The rationale for the age ranges selected for the temporal analysis is not clear. Have these been selected based on biological processes? How many samples and cells are in each age range? How many NSCs are in each age range? How many samples contribute to each cell type/group within these age ranges? Could differences in this explain some of the gene expression changes identified? I.e. if the predominant NPC subtype has changed with age, this could explain the changes in the NSC described.

We thank the reviewer for this comment. The gestational age ranges used in our temporal analysis were selected to reflect major developmental milestones: late first trimester (GW8–12), early second trimester (GW13–19), and late second trimester (GW20–26). These time windows align with critical transitions in cortical development, including progenitor expansion, neurogenesis, and early gliogenesis. This rationale is now clarified in the revised Methods section.

To minimize potential bias from uneven cell representation across ages or studies, we employed a pseudobulk strategy as described in the Methods. Gene expression was aggregated by donor and cell type (or time window), and differential expression was assessed using DESeq2, which models sample-level variance and controls for covariates. For identifying temporal patterns, we computed Pearson's correlation coefficients on pseudobulked expression matrices stratified by developmental window and donor.

Because each sample is treated as a single observation, the number of individual cells within a group does not influence statistical testing. Furthermore, our design ensures that each age group includes multiple donors and replicates across studies. This approach helps account for potential shifts in cell subtype composition, such as the relative abundance of NSC subtypes changing with age.

In summary, while we acknowledge that progenitor composition evolves over development, our analytic framework is designed to account for these changes and reduce sampling bias. We have updated the manuscript and supplementary material to better communicate this rationale and dataset structure. We agree that providing the number of samples and cells contributing to each age group and cell type improves transparency and interpretability. We have now included this information in new **Supplementary Fig. 4a** to address the reviewer's request.

7. It is not fully clear why some of the highlighted genes have been selected, with some previously published yet not cited well. TNC has already been shown to be a marker of basal radial glia (Pollen et al, 2024) and the LGALS3 related gene LGALS3BP has been linked to cortical expansion (Kyrrousi et al, 2021).

We thank the reviewer for this helpful comment. We are unsure which Pollen et al., 2024 paper is being referred to; however, we would like to note that Pollen et al., 2015 is cited in the original manuscript. Regarding LGALS3BP, while we agree that it is a particularly interesting protein with potential relevance to cortical expansion, we chose not to expand on every gene related to the marker genes highlighted in our study due to limitations in manuscript length and reference count. We have aimed to focus our discussion on genes most directly supported by our meta-analysis findings.

'While S100B is known to be expressed in OPCs and immature oligodendrocytes in the developing mouse brain^{68,69}, its expression dynamics in OPCs during human cortical development remain poorly understood.'

S100b has been studied in human brain development (reviewed by Hernández-Ortega et al, 2024), and has been described to be expressed in human OPCs and oligodendrocytes.

We thank the reviewer for this comment. We would like to note that the suggested citation (Hernández-Ortega et al., 2024) is already included in our original manuscript. While this review discusses *S100B* expression in the context of glial development, particularly oligodendrogenesis, it primarily references findings from mouse models. For example, under section "4.1. S100B role in glial development," the authors state that *S100B* expression is required for developing adult mouse brain oligodendrogenesis, citing three independent mouse studies.

Although *S100B* has been described in human OPCs and oligodendrocytes, its detailed expression dynamics during human cortical development remain less well defined. Our study contributes new transcriptomic insights into *S100B* expression in human OPCs and astrocytes during mid-gestation, helping to expand current understanding of its developmental regulation in the human brain.

8. The immunofluorescent images provided can be improved. In Fig 6, many cells appear to express LGALS3 but not Hopx (Fig 6d), including a strong band of staining in the marginal zone. Can the authors provide more detail on this? Why were the ages of tissue used selected? Does this align with the expression from the scRNAseq? The exact expression levels for LGALS3 and Hopx in these datasets should be included. The image quality of Fig.6g is very low and it is hard to see the expression clearly, however LGALS3 appears to be in the intermediate zone at 14 GW, not the progenitor zones. These images should include a progenitor marker to evaluate this, such as Sox2 or Hopx.

We appreciate the reviewer's comment highlighting LGALS3 expression in the marginal zone in **Fig. 6d**. As explained above, this region corresponds to the marginal zone, which includes resident microglia, which are established as prominent sources of LGALS3⁶. In addition, radial glial processes, particularly those extending from oRGs, could account for part of the LGALS3 signal detected in this region. Our analysis also revealed LGALS3 presence within the extracellular matrix. As a highly dynamic region enriched in signaling molecules and extracellular matrix components, the marginal zone forms a key transitional interface between the cortical surface and the meninges. Therefore, it is also plausible that the observed LGALS3 expression arises from paracrine contributions by neighboring cell types known to express this molecule, such as macrophages, meningeal fibroblasts, myeloid-derived cells, and microglia-like populations at the cortical boundary.

Regarding the choice of developmental timepoints, we selected GW16 in **Fig.6d** based on LGALS3's temporal expression profile in NSCs from the scRNA-seq analysis, where it begins to rise significantly during mid-gestation (see **Fig. 4a** and **Fig. 6f**). These timepoints are therefore appropriate for validating its expression in NSCs and oRGs during cortical development.

We agree with the reviewer that **Fig. 6g**, as originally submitted, was of limited resolution due to file size constraints imposed during figure upload for initial submission. To address this, we have replaced the figure with higher-resolution immunofluorescence images in the revised submission. Additionally, in response to the suggestion for a progenitor zone marker, we have now included new immunofluorescence images co-stained with SOX2 and GAL3 in **Fig.6g**, allowing more accurate evaluation of LGALS3 expression in relation to cortical progenitor zones.

9. Further validation for cell-type specific matrisome expression and temporal changes are needed to support the statements made. This could be via immunofluorescence or ISH for the top candidates. In Fig. 4b, as the major changes described occur during the early second trimester, it would be interesting to see how expression levels of each matrisome component highlighted change in each age examined to understand the dynamics of this matrisome increase. In Fig. 4d, the GO analysis for general terms could also be complemented with analysis of specific ECM functions (similar to that shown in Fig 1a).

We thank the reviewer for this insightful comment. While we agree that experimental validation such as immunofluorescence or in situ hybridization of individual matrisome genes would further strengthen the mechanistic interpretation of our findings, the primary aim of this study is computational hypothesis generation through the systematic integration and analysis of existing scRNA-seq datasets.

Our analyses provide robust observational evidence for cell type and temporally specific matrisome expression signatures across six independent studies and 37 fetal donors, using methods such as pseudobulk differential expression, gene co-expression networks, and correlation-based modeling. These findings are also consistent with established principles of human cortical development, particularly during the early to mid-second trimester.

While we agree that examining age-specific expression dynamics of individual matrisome components and expanding the GO analysis to include specific ECM-related functions would provide further insight, these analyses are beyond the current scope of this study. Our primary aim is to provide a systematic and integrative overview of matrisome expression during human cortical development. To support further exploration by readers, we have included comprehensive gene-level expression data across developmental time points in **Supplementary Table 6**, which allows users to query genes or stages of particular interest.

10. 'Non-neuronal lineage cell types exhibit the strongest tendency toward positive temporal regulation, with endothelial cells showing an 85.0% (17 out of 20 genes) and OPCs displaying 80.6% (25 out of 31 genes) of temporally increasing matrisome genes, suggesting their growing contribution to matrisome-mediated functions during cortical development'

The authors do not comment on the type of matrisome genes expressed by endothelial cells. These cells produce a vast amount of vascular basement membrane as blood vessels develop, and it is not surprising that this increases over the course of development, as blood vessels grow and mature and the basement matures with them.

We thank the reviewer for this comment. We agree that endothelial cells contribute significantly to the vascular basement membrane, and that the observed increase in matrisome gene expression in these cells over developmental time likely reflects ongoing vascular growth, maturation, and ECM remodeling, all of which are essential for establishing the cerebral vasculature.

In our temporal analysis, many of the matrisome genes upregulated in endothelial cells are indeed consistent with basement membrane components and ECM-associated structural proteins involved in angiogenesis and vessel stabilization (**Supplementary Table 5**). These include genes such as *COL4A6*, *PRELP*, *PCOLCE*, and *NID1*, which are known to play roles in vascular basement membrane assembly¹⁵⁻¹⁸.

We have now expanded the relevant section in the manuscript to briefly discuss the nature of the matrisome genes expressed by endothelial cells and to contextualize their temporal upregulation in relation to vascular development.

11. No validation has been provided for the ligand-receptor interactions proposed from the CCI analysis. Immunofluorescence data are needed to support these statements, along with any citations available to show that these pathways are active in the cell types listed.

We thank the reviewer for this important comment. We acknowledge that experimental validation would provide additional support for the ligand-receptor interactions inferred from our cell-cell interaction analysis. Due to the secreted nature of many ligands and the often transient or spatially diffuse nature of cell–cell signaling, immunofluorescence is not always suitable for confirming these interactions at the protein level. Moreover, our study is designed primarily as a computational meta-analysis aimed at generating testable hypotheses regarding matrisome-mediated signaling pathways during human cortical development.

The ligand-receptor interactions were inferred using CellChat, a well-established computational framework that predicts statistically significant intercellular signaling pathways based on co-expression of ligands and receptors in defined cell populations. While these predictions are not direct evidence of pathway activity, they offer an unbiased and data-driven method to identify potential matrisome-mediated communication events.

We agree with the reviewer that citing prior studies supporting the biological relevance of these pathways strengthens the interpretation. In our original manuscript, we describe representative matrisome-related signaling pathways identified in our analysis. The PTN pathway, known for its role in modulating proliferation, differentiation, and cell survival^{19,20}, exhibited robust bidirectional communication between neural and glial cells, with signaling notably converging on tRG and astrocytes. The MK signaling network, which supports cell growth and neural plasticity²¹, showed a more focused interaction pattern, primarily targeting tRG. In contrast, SEMA6 signaling, associated with neuronal morphogenesis and migration^{22,23}, revealed distinct interactions targeting neuronal populations.

While experimental validation of all predicted interactions is beyond the scope of this study, we emphasize that our goal is to provide a resource and prioritization framework for future experimental work. We have revised the text to clarify this positioning and added citations where applicable to support the biological plausibility of key interactions.

12. 'Astrocytes, M.GExN, and NSCs had the highest number of temporal marker genes, suggesting significant contribution of matrisome gene expression during development.'

Is this due to fact that the numbers/proportions of these cells change greatly over 8-26gw?

We thank the reviewer for this thoughtful comment. The statement refers to data presented in **Supplementary Fig. S4** and **Supplementary Table 6**, which show that astrocytes, M.GExN, and NSCs exhibit the highest number of temporal matrisome marker genes across the developmental time windows analyzed.

It is highly unlikely that this observation is simply due to differences in cell numbers or sample representation across ages. Importantly, the identification of matrisome marker genes was based on within-cell type comparisons across time windows, not between different cell types. Therefore, the relative number of samples or cells per cell type does not influence the detection of marker genes under our analytical framework.

As described in the Methods and explained further above, we employed a pseudobulk approach and applied statistical models (DESeq2 and Pearson's correlation) that control sample-level variability and minimize noise from unequal sampling. These methods are specifically designed to reduce the influence of sample size disparities. Moreover, if sample size were a dominant factor, we would not expect to see the highest number of temporal

marker genes in the early second trimester, where sample numbers tend to be lower than in the late first or late second trimester (**New Supplementary Fig. 4a**).

Thus, the observed enrichment of temporal marker genes in astrocytes, M.GExN, and NSCs reflects true biological dynamics of matrisome gene expression during cortical development, rather than sampling artifacts. These findings suggest a more active and temporally regulated role for matrisome components in the developmental progression of these cell types.

13. Is the pseudotime analysis performed based solely on the authors own clustering? Temporal information is available for each sample analysed, this should also be used for an in depth analysis of gene expression changes over time for LGALS3.

We thank the reviewer for this comment. We would like to clarify that pseudotime analysis is not used in our study to assess temporal gene expression patterns, but rather to infer developmental trajectories and predict the relative positioning of LGALS3⁺ cells along these trajectories. The pseudotime analysis was performed using our own cell type annotations and clustering to model lineage progression, specifically focusing on the developmental fate of LGALS3⁺HOPX^{high} cells.

To address temporal expression dynamics, we used a separate, dedicated approach. In **Fig. 4a**, we present the temporal expression pattern of LGALS3 in NSCs using a custom function that aggregates gene expression at the donor level and applies linear regression and Pearson's correlation to quantify expression changes across gestational age. This method leverages the actual temporal metadata available for each sample and provides a robust measure of LGALS3 dynamics over developmental time.

We have clarified the distinction between pseudotime inference and temporal modeling in the revised manuscript to avoid confusion and to better reflect the complementary nature of these analyses.

Minor comments:

1. The figure legend from Fig. S2 is incomplete (missing 2c-e).

We appreciate the reviewer pointing this out. We have now revised the legend to include complete descriptions for all panels, including the methods and interpretations relevant to **Supplementary Fig. 2c–2e**.

2. Details of the statistical tests in Fig. 6e, 7f are missing (t-tests?)

We thank the reviewer for this comment. The panels in question present the proportions of HOPX⁺GAL3⁺ and HOPX⁺GAL3⁻ cells, rather than comparing means between independent experimental groups. Therefore, we did not perform formal statistical testing, as the intent of these plots is to illustrate distribution patterns across donors and samples and to highlight relative proportions, rather than to assess statistical significance.

3. Fig. 6d lacks a box to show where d' is located.

The ECM is an essential component for multi-cellular organisms. In this manuscript, Gim et al. utilized a meta-analysis of scRNA-seq datasets of human cortical development to systematically characterize gene expression signatures related to matrisome. With this analysis, the authors found matrisome gene expression showing both cell-type and developmental stage-specific features, highlighting the important role of ECM in cortical development and diseases. Overall, the concept of this work is meaningful, and the analysis is rigorous. Below are some suggestions for the authors before the manuscript is considered for publication.

We sincerely thank the reviewer for their thoughtful summary and positive evaluation of our work. We are pleased that the reviewer finds the concept meaningful and the analysis rigorous.

1. Since the overlapping of matrisome genes and NDD risk genes is a major delivery, Fig. 1b should perform a significant test.

We thank the reviewer for this helpful suggestion. We agree that a statistical test is important to support the observed overlap between matrisome genes and NDD risk genes. In response, we have now performed a chi-square test to assess the significance of this association and have incorporated the results into **Fig. 1b**. This addition strengthens the evidence for enrichment of NDD risk genes within the matrisome and has been described in the revised figure legend and Methods section.

2. The authors extensively discuss the existence of cell-type or temporal-specific features of matrisome genes. However, the underlying gene regulation mechanisms were not discussed. Fig. 6k identified potential TFs driving LGALS3 expression. Since all the data come from scRNA-seq, it will be worth to check whether there is a co-expression significance between LGALS3 and its potential TFs. The same analysis may apply to other matrisome genes to imply an upstream gene regulation.

We thank the reviewer for this insightful comment. As noted, our study focuses on characterizing the cell type- and temporally specific features of matrisome gene expression during human cortical development. We agree that investigating upstream gene regulatory mechanisms adds valuable context and mechanistic insight.

To address this, we identified transcription factors (TFs) that may regulate LGALS3 expression using a multi-step approach:

- (1) generating pseudotime-ordered expression profiles of all genes,
- (2) applying K-means clustering to group genes with similar pseudotemporal dynamics,
- (3) identifying co-expressed genes using Pearson correlation,
- (4) extracting transcription factors within the same cluster as LGALS3, and
- (5) verifying TF binding motifs near the LGALS3 transcription start site.

Because steps (2) and (3) ensure that candidate TFs share both expression patterns and co-regulation with LGALS3, this approach provides strong evidence for significant co-expression. In response to the reviewer's comment, we have revised the manuscript to clarify this analytic framework and highlight the co-expression findings, including TFs such as *SOX9*, *HES1*, and *CEBPB*.

While the application of this method to all matrisome genes is beyond the current scope, we agree this is an important direction for future research. To support such efforts, we have made our custom code and functions publicly accessible, and we hope they will aid future studies aiming to systematically dissect the regulatory networks of additional matrisome genes.

We appreciate the constructive suggestions provided. We believe the revisions have strengthened the clarity and interpretability of our findings

Reviewer #4 (Remarks to the Author):

Reviewer #5 (Remarks to the Author):

References

- 1 Amin, S. & Borrell, V. The Extracellular Matrix in the Evolution of Cortical Development and Folding. *Front Cell Dev Biol* **8**, 604448 (2020). <https://doi.org/10.3389/fcell.2020.604448>
- 2 Pokhilko, A. *et al.* Global proteomic analysis of extracellular matrix in mouse and human brain highlights relevance to cerebrovascular disease. *J Cereb Blood Flow Metab* **41**, 2423-2438 (2021). <https://doi.org/10.1177/0271678X211004307>
- 3 Naba, A. *et al.* The extracellular matrix: Tools and insights for the "omics" era. *Matrix Biol* **49**, 10-24 (2016). <https://doi.org/10.1016/j.matbio.2015.06.003>
- 4 Shao, X., Taha, I. N., Clauser, K. R., Gao, Y. T. & Naba, A. MatrisomeDB: the ECM-protein knowledge database. *Nucleic Acids Res* **48**, D1136-D1144 (2020). <https://doi.org/10.1093/nar/gkz849>
- 5 Vinsland, E. & Linnarsson, S. Single-cell RNA-sequencing of mammalian brain development: insights and future directions. *Development* **149** (2022). <https://doi.org/10.1242/dev.200180>
- 6 Garcia-Revilla, J. *et al.* Galectin-3, a rising star in modulating microglia activation under conditions of neurodegeneration. *Cell Death Dis* **13**, 628 (2022). <https://doi.org/10.1038/s41419-022-05058-3>
- 7 Bellac, C. L., Coimbra, R. S., Simon, F., Imboden, H. & Leib, S. L. Gene and protein expression of galectin-3 and galectin-9 in experimental pneumococcal meningitis. *Neurobiol Dis* **28**, 175-183 (2007). <https://doi.org/10.1016/j.nbd.2007.07.005>
- 8 Hellstrom Erkenstam, N. *et al.* Temporal Characterization of Microglia/Macrophage Phenotypes in a Mouse Model of Neonatal Hypoxic-Ischemic Brain Injury. *Front Cell Neurosci* **10**, 286 (2016). <https://doi.org/10.3389/fncel.2016.00286>
- 9 Mehina, E. M. F. *et al.* Invasion of phagocytic Galectin 3 expressing macrophages in the diabetic brain disrupts vascular repair. *Sci Adv* **7** (2021). <https://doi.org/10.1126/sciadv.abg2712>

- 10 Rotshenker, S. Galectin-3 (MAC-2) controls phagocytosis and macropinocytosis through intracellular and extracellular mechanisms. *Front Cell Neurosci* **16**, 949079 (2022). <https://doi.org/10.3389/fncel.2022.949079>
- 11 Sideris-Lampretsas, G. *et al.* Galectin-3 activates spinal microglia to induce inflammatory nociception in wild type but not in mice modelling Alzheimer's disease. *Nat Commun* **14**, 3579 (2023). <https://doi.org/10.1038/s41467-023-39077-1>
- 12 Pollen, A. A. *et al.* Molecular identity of human outer radial glia during cortical development. *Cell* **163**, 55-67 (2015). <https://doi.org/10.1016/j.cell.2015.09.004>
- 13 Florio, M. *et al.* Human-specific gene ARHGAP11B promotes basal progenitor amplification and neocortex expansion. *Science* **347**, 1465-1470 (2015). <https://doi.org/10.1126/science.aaa1975>
- 14 Torres-Berrio, A., Hernandez, G., Nestler, E. J. & Flores, C. The Netrin-1/DCC Guidance Cue Pathway as a Molecular Target in Depression: Translational Evidence. *Biol Psychiatry* **88**, 611-624 (2020). <https://doi.org/10.1016/j.biopsych.2020.04.025>
- 15 Khoshnoodi, J., Pedchenko, V. & Hudson, B. G. Mammalian collagen IV. *Microsc Res Tech* **71**, 357-370 (2008). <https://doi.org/10.1002/jemt.20564>
- 16 Kalluri, R. Basement membranes: structure, assembly and role in tumour angiogenesis. *Nat Rev Cancer* **3**, 422-433 (2003). <https://doi.org/10.1038/nrc1094>
- 17 Bengtsson, E., Neame, P. J., Heinegard, D. & Sommarin, Y. The primary structure of a basic leucine-rich repeat protein, PRELP, found in connective tissues. *J Biol Chem* **270**, 25639-25644 (1995). <https://doi.org/10.1074/jbc.270.43.25639>
- 18 Rangan, R., Sad do Valle, R. & Tovar-Vidales, T. Expression of procollagen C-proteinase enhancer 1 in human trabecular meshwork tissues and cells. *Exp Eye Res* **225**, 109280 (2022). <https://doi.org/10.1016/j.exer.2022.109280>
- 19 Pufe, T., Bartscher, M., Petersen, W., Tillmann, B. & Mentlein, R. Expression of pleiotrophin, an embryonic growth and differentiation factor, in rheumatoid arthritis. *Arthritis Rheum* **48**, 660-667 (2003). <https://doi.org/10.1002/art.10839>
- 20 Herradon, G. & Perez-Garcia, C. Targeting midkine and pleiotrophin signalling pathways in addiction and neurodegenerative disorders: recent progress and perspectives. *Br J Pharmacol* **171**, 837-848 (2014). <https://doi.org/10.1111/bph.12312>
- 21 Yildirim, B., Kulak, K. & Bilir, A. Midkine: A Cancer Biomarker Candidate and Innovative Therapeutic Approaches. *Eur J Breast Health* **20**, 167-177 (2024). <https://doi.org/10.4274/ejbh.galenos.2024.2024-4-7>
- 22 Perez-Branguli, F. *et al.* Reverse Signaling by Semaphorin-6A Regulates Cellular Aggregation and Neuronal Morphology. *PLoS One* **11**, e0158686 (2016). <https://doi.org/10.1371/journal.pone.0158686>
- 23 Kerjan, G. *et al.* The transmembrane semaphorin Sema6A controls cerebellar granule cell migration. *Nat Neurosci* **8**, 1516-1524 (2005). <https://doi.org/10.1038/nn1555>